# Factors that Impact Farmers' Organic Conversion Decisions

**Philippos Karipidis** [1,*] and **Sotiria Karypidou** [2]

1   Department of Agriculture, International Hellenic University, 57001 Thessaloniki, Greece
2   Computer Science, International Hellenic University, 65404 Kavala, Greece; sokaryp@cs.ihu.gr
*   Correspondence: philika@ihu.gr or siphn@yahoo.gr

**Abstract:** This article helps to answer the question of how the diffusion of organic farming could be accelerated by analyzing farmers' decisions. Given the fragmentation of the research findings, the determinants of farmers' organic conversion decisions were integrated into a framework that enables a holistic approach to be adopted in research and policy scheduling. The most important factors of the external farm environment are the organic product demand, product price, access to markets, available technologies, education, knowledge transfer, peer networks, society's attitudes, and subsidy provision. The most important farm characteristics are the farm's location, farm size, enterprise, expected costs, profits, knowledge, information and communication technology use, farmers' age, education, gender, off-farm activities, attitudes, and beliefs regarding organic farming and willingness to preserve the environment. Of particular importance are farmers' satisfaction with economic incentives, the perception of technical problems, and the certification process. Such comprehensive information enables public authorities to bring about changes in the most important factors that effectively accelerate organic conversion decisions and to assess policy implementation. The market participants are facilitated to implement eco-strategies by encouraging farmers to decide to convert. Future research should broaden the sets of factors that are explored, taking into consideration the interactions and time-dependent changes that exist.

**Keywords:** organic farming; conversion decisions; determinants of acceleration; sets of factors; market; policy

## 1. Introduction

Because agriculture is a key source of environmental pressures, the need to urgently reduce the impacts of agricultural activities on biodiversity, freshwater and marine pollution, greenhouse gas and ammonia emissions, and soils [1] has been recognized by the European Union. Thus, three of the post-2020 Common Agricultural Policy (CAP) objectives concern the environment and climate change. More specifically, a substantial contribution is scheduled to mitigate climate change, foster sustainable development and efficient management of natural resources, protect biodiversity, enhance ecosystem services, and preserve wildlife habitats and landscapes [2]. These challenges can be addressed by the adoption of more sustainable agricultural production systems such as organic farming [3–14]. This was taken into account during the new CAP formation, and the achievement of a target of 25% organic farmland by 2030 was decided.

### 1.1. Specification of the Benefits of Organic Farming

Some researchers highlight the multiple benefits of organic farming. For example, Horrillo et al. [5] found that greenhouse emissions in four types of organic livestock farming are lower than those from conventional farms, while the levels of carbon sequestration are noticeably higher. Cattell Noll et al. [6] note that organic crop and animal production can reduce global nitrogen pollution, while Borsato et al. [7] report that organic viticulture can be applied without incurring economic losses and with better preservation of natural capital. The environmental benefits of organic viticulture are linked with the water and carbon

footprints, pesticide and fertilizer management, organic matter content, soil compaction and erosion, and landscape quality. A more precise quantification of the total environmental consequences was conducted by Zaher et al. [8]. They found that the ecotoxicity effects per kilogram of potato production were reduced by approximately three orders of magnitude in a US organic case study farm as compared to average conventional production.

Apart from the above mentioned benefits, organic agriculture can promote more compassionate treatment of animals, while it can also provide important benefits to human health [9]. For example, Welsh et al. [10] note that contaminant levels in food, such as antibiotics and pesticides, were undetectable in organic samples but were identified in conventionally produced milk samples, with multiple samples exceeding the federal limits. In addition to this, synthetic growth hormones were also detectable in conventionally produced milk. Though the scientific evidence remains scarce [11], some recently published studies highlight [9,12] that organic food seems to be healthier compared to conventional food due to the fact of its higher content of bioactive compounds and polyunsaturated fatty acids. Moreover, organic food has lower cadmium content and other unhealthy substances, such as pesticide residues, which are linked with gut microbiota dysbiosis, immune-related disorders, toxicity in humans, and negative impacts on cognitive development in children.

Some indirect effects can be also attributed to organic products such as the health outcomes closely linked to the eating habits of organic consumers. For example, the diet of organic consumers is often richer in fruits, vegetables, legumes, and whole grains and lower in meat intake. Such a dietary pattern leads to a lower incidence of metabolic diseases such as cardiovascular diseases and diabetes mellitus type 2. In addition, the greater content of dietetic fiber in organic food may have a positive effect on gut microbiota and health, because the risks for some diseases and allergies are reduced [11], and it also demonstrates a potential beneficial effect on obesity among adults. It is also important to note that the diet of organic consumers, which involves less animal-based food products, has an indirect environmental benefits. Such a diet enables the carbon footprint and land use to be further reduced [13,14].

### 1.2. The Organic Conversion

Given the benefits organic farming creates, it could be argued that organic food supply is a response to the European policy target for the environmental impact of agriculture to be reduced [1,2]. This enables certain environmental consequences to be minimized and, thus, agriculture to contribute to the 12th Sustainable Development Goal (UN) of responsible consumption and production [15] being achieved. Taking into account the above presented benefits, the more widespread organic food production and consumption is, the higher the contribution of the agro-food sector to achieving this goal.

### 1.2.1. The Diffusion of Organic Farming

Selected statistical data [16] indicate that organic farmland increased by 65% between 2008 and 2018 in the EU, with organic farmland shares ranging among countries. Organic livestock also increased during the same period. For example, the populations of organic sheep and poultry increased by 67.8% and 127.9%, respectively. In regard to total organic food sales, it was noticed that they increased by 121% in the EU, reaching EUR 37.4 billion (EUR 97 million globally). Although the rates of diffusion appear to be high, the spread of organic farming cannot be considered satisfactory, because only 7.7% of the EU's farmland was organic on 2018. Thus, the question arises of "how will organic farming expand from this low level to 25% in 2030" as required by European policy [2]. The question becomes even more important for the whole Earth, where only 1.5% of the world's farmland was organic in 2018 (71.5 million hectares), while more than double the environmental footprint is expected by 2050, because food consumption is expected to double. The urgency of a change becomes further apparent, because any increase in food expenditure by 1% is followed by an increase of 1.4% in the environmental footprint [1,16,17].

The phase of production has the greatest negative environmental effect among the sub-sectors of the agro-food supply system. Its greenhouse gas emissions contribute to about 65% to 85% of the total system's emissions [18]. Thus, a great reduction in the total environmental consequences of food supply can be achieved if the number of organic farmers as well as the diffusion of organic farming activities substantially increase worldwide. Following the EU policy target [2], such a conversion should lead to at least a tripling of the organic farmland by 2030. Given that the organic farmland increased by only 65% in the last decade, it is questionable whether this goal can be achieved.

### 1.2.2. Farmers' Conversion Decisions

The economic, environmental, and social issues of organic farming, combined with the low level of its diffusion, the heterogeneity of conditions worldwide, and the social and policy interest have encouraged many researchers to explore the factors that can affect decisions made by farmers related to the conversion of conventional farming activities to organic [19–70]. Each of the studies identified a limited number of factors, while few of them extended the exploration in more than one country [53]. Such a fragmentation of the knowledge does not enable sufficient information to be provided to public authorities aiming at the diffusion of organic farming to be accelerated. A study that integrates as many factors as possible, which impact on farmers' decisions to convert worldwide, into a framework is a challenge. The present article aimed to integrate the numerous factors that determine farmers' decisions in regards to the organic conversion worldwide into a framework.

This framework could facilitate a more holistic approach to be adopted in the empirical research, while it could also provide comprehensive information that enables public authorities to choose between several sets of factors, the most important of which will lead to achieving the highest effectiveness and efficiency in policy implementation. This information can also help farmers' cooperatives, food producers, and marketers operating in a totally globalized environment to effectively implement an eco-friendly strategy by properly encouraging farmers to proceed with organic conversion. In addition, because the environmental problems are spreading beyond a narrow geographical area or country, such a framework enables environmental policy to be simultaneously implemented in more regions, in groups of countries (such as the EU), across continents, and worldwide. Because it cannot be achieved by empirical research that focuses on a limited number of factors, the present paper is based on published research to identify, bring together, and systematically integrate into a framework the factors that impact the organic conversion decisions of farmers worldwide.

## 2. Methodology

The present review is conducted in an integrative way, following Torraco's [71] definitions and recommendations. Such an "integrative review" focuses on ideas and results extracted from the individual papers, only a few details of which are provided. More specifically, we review and synthesize selected literature that focuses on the determinants of the decisions of farmers to convert their conventional farming activities into organic activities. Selected studies are cited as examples of results illustrating the study's argument, which is linked with the fragmentation and insufficiency of information that previous studies provide to public authorities and private actors who aim to accelerate the conversion of conventional farming into organic farming. Based on selected results of the extracted papers, a summary of milestones in the research area is synthesized [71–73], enabling a more holistic understanding of the determinants of farmers' conversion decisions.

To understand how the farmers' organic conversion decisions are influenced by various factors, in a first step, we searched for a theoretical framework. The process to build it is presented in Section 3. Along with this, the search strategy has been designed. The criteria for the selection of articles from the literature have been defined in a way that enables us to select all articles that explored the determinants of farmers' organic conversion

decisions worldwide. Titles and abstracts of the articles were reviewed to verify these criteria. If the inclusion requirements were met or if this remained unclear, the articles were fully read. In cases where the full text revealed that farmers' organic conversion decision determinants were not present, the paper was not included in the core articles. From the articles retrieved, additional references were identified by a manual search among the cited references. The Google Scholar aggregator was initially used for a broad search for studies published during the last two decades. Such a narrowing of the search enabled the potential biases that may be due to the lack of experiences related to organic farming and organic markets to be minimized. We used the terms "farmers' decisions", "farmers' attitudes", "farmers' perceptions", "farmers' intentions", and "conversion decisions" in combination with "organic farming", "organic agriculture", "organic farmers", and "organic food" in the search. The first filtration was based both on the titles of the papers and whether they were published in peer-reviewed journals. A preference was indicated towards articles that were published in journals that have broad academic recognition in several scientific subjects including the fields of agricultural economics, management, and marketing. In only a few cases were other scientific publications complementarily used. After the first short reading, we chose the articles that mainly explore the determinants of farmers' decisions, intentions, attitudes, and perceptions related to organic conversion.

As is above noted, after the initial review of each abstract, we proceeded with an in-depth review to identify decision determinants. Although large sets of factors have been previously explored [19–70], we chose the factors that were found to affect farmers' conversion decisions. Of the numerous articles included in this study, 53 references were core articles that included one or more determinants of farmers' behavior that was related to organic conversion. Then, the theoretical framework was developed (Section 3) that links farmers' organic conversion decisions with the external farm business environment [74–76]. The determinants of farmers' organic conversion decisions identified worldwide were next integrated into the framework, and organized in sets and sub-sets of factors. Definitions and discussion with regard to the effect for each set and sub-set of factors follow in Sections 4–6. The main steps we followed are graphically shown in Appendix A, Figure A1.

## 3. Organic Conversion Decision Framework

The theory of planned behavior allows for the identification of the determinants of individual behavioral intentions. This has been adopted in some studies analyzing farmers' choices in regard to organic conversion [36,57,58,66,77]. Feola and Binder [74], in an attempt to link farmers' behavior with agricultural systems, built an agent-centered framework that supports the understanding of farmers' behavior consistently with the perspective of agricultural systems as complex social–ecological systems. In a later published study, Schlüter et al. [75] provided a framework (Modelling Human Behavior) to facilitate a broader inclusion of theories on human decision-making in formal natural resource management models. Its entities and processes have been clearly defined by Zagaria et al. [76], who modeled the transformational adaptation to climate change. More precisely, three principal entities were outlined: an external social and biophysical environment within which agents make decisions; individual agents with their goals, values, and assets; and a set of perceived behavioral options that agents may choose to perform. These entities interact through three consecutive processes representing adaptation decision-making.

We adopted the framework of Zagaria et al. [76], because it enables the numerous sets of factors that have previously been examined to be integrated into a common theoretical foundation. Following this framework, farmers first update their characteristics based on their perception of changes to the external farm environment; they then select to convert conventional farming activity to organic activity based on its capacity to meet their goals. Lastly, they implement the selected organic conversion with repercussion to internal and external characteristics. Thus, internal and external variables directly determine the adoption and implementation. These variables can be distinguished into two groups. In the first group, the factors of the external farm business environment are included. Changes to

these factors can be caused by public interventions [17,31,55,68], but farmers cannot control them in the short term. The second group includes all of the other factors that constitute the internal farm business environment. Changes in these factors can be caused both by the farmers themselves and by the policy. The framework we built is shown in Figure 1, while the literature references on which the creation of the sets of factors was based can be found in Appendix A, Table A1.

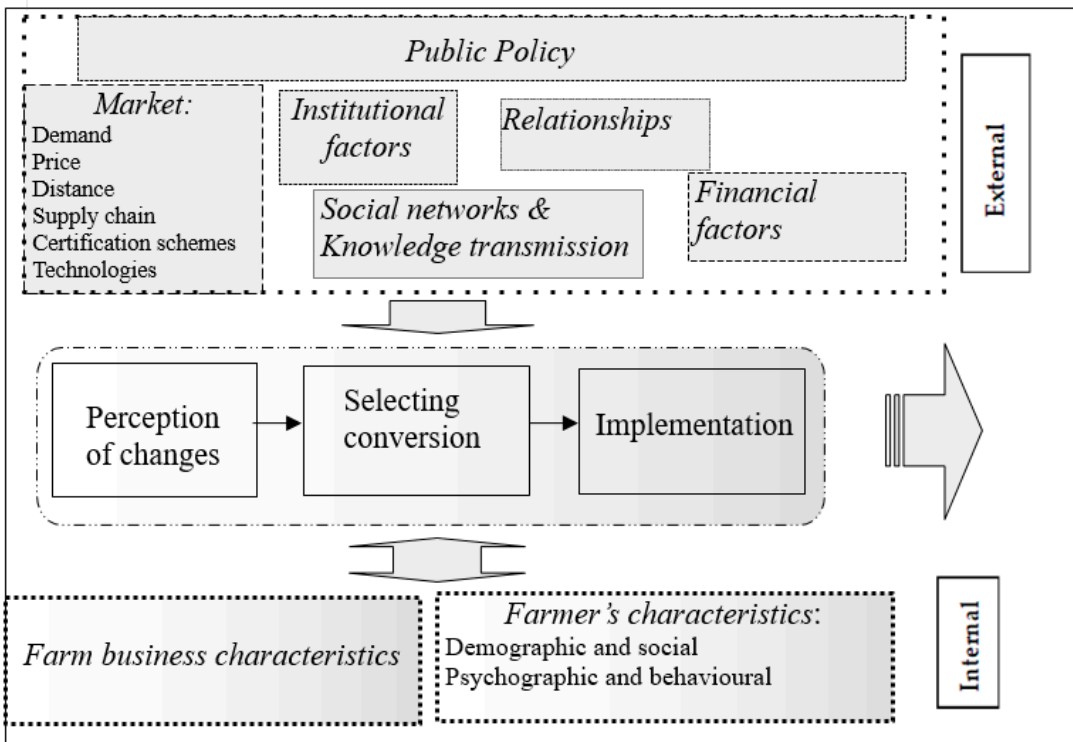

**Figure 1.** Farmers' organic conversion decisions framework. Sets/sub-sets of factors effect farmers' decisions to convert conventional farming activities into organic activities.

Each of the core literature references (53) provided support for one or more the sets/sub-sets of factors that were formulated. Table 1 presents the number of references and the corresponding sets/sub-sets of factors to which these references provide support. As is shown, two of the references provide support for 7 or 8 sets and sub-sets of factors, while sixteen references provide support for one set or sub-set of factors. Each of these references is cited one or more times in Sections 4 and 5, reflecting an equal number of factors that impact on farmers' organic conversion decisions. Table 2 presents the number of references in accordance with the citations made in Sections 4 and 5. As it is shown, two of the articles are cited 10 to 16 times, while fifteen of them are cited one time.

**Table 1.** References in accordance with the sets of factors.

| Number of References | Sets of Factors |
| --- | --- |
| 2 | 7–8 |
| 8 | 5–6 |
| 12 | 3–4 |
| 15 | 2 |
| 16 | 1 |

**Table 2.** References in accordance with the citations.

| Number of References | Citations |
|:---:|:---:|
| 2 | 10–16 |
| 6 | 8–9 |
| 4 | 6–7 |
| 13 | 4–5 |
| 13 | 2–3 |
| 15 | 1 |

## 4. External Farm Business Environment

The factors identified were related with the market, the institutional environment, social networks and knowledge transfer, the relationships of market participants and institutions, the financial environment, and public policy.

### 4.1. Market Factors

Availability of markets for organic products was included among the main reasons for farmers' willingness to convert [25]. The global organic food and beverage market is rapidly increasing, reaching EUR 97 million euros in the period 1990–2018. North America has the largest market with 45% of total organic sales, while Europe follows with 42% of total sales. The average per capita consumption reaches EUR 12.9, substantially ranging among continents from 119.9 in North America to 50.5 in Europe and 0.01 in Africa. There also exist substantial differences among countries, such as Denmark and Switzerland, which have the highest per capita organic consumption (about EUR 312), while some countries in Africa and Asia have zero consumption [16]. These data indicatively reflect the different conditions that farm businesses face as well as the market-related aspects that can impact on farmers' conversion decisions worldwide. The factors that were identified were organized into six sub-sets, taking into account their importance as it was revealed in the published research [33–36]: the demand, the distance to a point of sale, the price of the organic product, the supply chain, the certification systems, and the technologies.

#### 4.1.1. Demand

McCullough et al. [19] note that global shifts in consumption, marketing, production, and trade are leading to organizational changes along the food chain. The market demand for organic produce, as has been previously noted, has rapidly been increasing, thus determining conversion decisions, because this is the main influencing factor that producers take into consideration to proceed with organic production [20–22]. This is also confirmed by Kleemann et al. [37], who studied the global food market demand. Thus, the higher the demand, the higher economic value the business model creates, which positively impacts organic conversion decisions. As a result, environmental value is created. In addition, social value is created, because consumers can find a richer product variety, and they are encouraged to follow a healthier diet. Although demand is one of the main determinants of a farmers' decisions, its importance substantially varies depending on circumstances such as country, location, and farm size [31,34,35,64].

#### 4.1.2. Price

Because conversion costs in organic farming are high [78], the price of organic food is a crucial factor influencing conversion decisions [23,25,70]. Organic price premiums are powerful instruments to motivate the adoption of organic techniques [26,27]. In an earlier study, it is estimated that in a scenario where there are no adoption costs, a price premium of 40% leads to organic adoption of 37% of farms, while a 90% premium may trigger the adoption of 70% of farms [28]. In a more recently published study, it was reported that

organic-certified farming yields a significantly higher return on investment (ROI) than GlobalGAP-certified farms, mainly due to the price premium on the organic market [37].

These findings highlight the product price of the most important determinants of farmers' conversion decisions. The higher prices help a higher economic value to be created. This enables organic producers to enhance economic performance and, subsequently, to choose organic conversion, which also allows environmental and social value to be created due to the richer product variety offered on the market and the improved safety of the consumers' organic products. Thus, sustainable entrepreneurs can balance the economic, environmental, and social value and subsequently contribute to achieving the goal of sustainable production and consumption.

### 4.1.3. Distance to the Market or Point of Sale

Some research results reveal the importance of the access to the market and distance between the food producing location and the market or point of sale, for example, the distance to an urban center or market [23,29,35], a product-cleaning location [30], a certified-organic abattoir [23], or the available organic markets [31]. The lack of supply and delivery points within an acceptable travel distance [32], which is the long distance between the food production location and selling points, negatively impacts on organic conversion decisions. The importance of the distance depends on the circumstances such as the product, place of production, food market, and farm's characteristics. For instance, it can be higher when the product is more perishable and bulkier, the farm businesses are physically isolated and smaller, or the marketing system is more effective [24,34,35,37].

### 4.1.4. Supply Chain

The supply chain constitutes a network of participants and represents the steps it takes to obtain the organic product from its original state to the final customer. It includes different people, entities, information, resources, and activities and enables farmers to create economic, environmental, and social value, impacting their decisions to convert their activities to organic activities. More specifically, the supply chain can reduce transaction costs and liability risks associated with retail efforts to govern product sustainability issues in the upstream (farmers) side [38]. This finding is confirmed by Vallandingham et al. [78], highlighting that the high efficiency that grocery retail supply chains have achieved over the last years has led to cost-focused supply chains that are able to deliver high volumes of products at low prices. Furthermore, the reverse information flow that the supply chain enables to be conducted supports organic conversion decisions [39]. The same stands for the finance and contract design of food buyers, which assists farmers as well as for the support provided to farmers for the documentation of environmental services. It was found that these aspects positively influenced farmers' conversion decisions [72].

The connection of organic farm businesses with the domestic or world markets is an aspect of the supply chain that can impact conversion decisions. More specifically, Brindley and Oxborrow [39] note the importance of intermediaries and relationships as well as the pushing and promoting of sustainable products into core markets. The strong trade relationship with customers, such as in the soybean supply in Europe, facilitates an upgrading of the supply chain. It enables new opportunities to be created, because it allows farmers to differentiate their products based on environmental quality in order to access price premiums in niche markets. These issues in combination with the supply chain structure help us to understand why the farmers of a certain country have adopted environmental certification programs on a larger scale than in other countries [40]. In addition to the above, global shifts in consumption, marketing, production, and trade are leading to organizational changes along the food supply chain, thus determining its impact on conversion decisions [40].

### 4.1.5. Certification Schemes

The availability of certification schemes in the market, the certification-related costs, and the certification process can affect organic conversion decisions. More specifically, the existence of alternative sustainability certification schemes is important for conversion decisions [38,41,79], while the absence or underdevelopment of sustainability certification schemes constrains the retailers' ability to exercise environmental and social responsibility in the upstream supply chain [22].

Because certification by a third party contributes to the elimination of food buying uncertainty, organic certification can be a decisive issue for customers as well as for farmers because it can be profitable [37]. Thus, the certification cost, which is charged to producers and consumers in combination with the certification process, is important for organic decisions [31], given that the cost of certification is high and the certification process is confusing. These can be partially attributed to the expensive organic certification procedures as well as to the long duration of the agronomic experimentation and seed multiplication [42]. This was also confirmed by Uematsu and Mishra [43], who explained that certified-organic producers spend significantly more on labor, insurance, and marketing charges than conventional farmers, and Bravo et al. [44] and Siepmann and Nicholas [26], who note that the perceived bureaucracy associated with organic certification negatively affects farmers' expectations and decisions. In regard to the perceived reliability of organic certification, it was not found to affect food producers' satisfaction [44].

### 4.1.6. Technologies

The role of technological innovations has been broadly recognized in organic conversion [30,33,36]. Such innovations are linked with input materials, information and communication technology (ICT) applications, systems of irrigation, crop protection, animal feeding, crop and breed management, information acquisition, food quality management and traceability, marketing, etc. Because the use of ICT in food supply has rapidly developed over the last years, becoming a promising issue, we chose to rely on selected literature to understand the expected impact of available ICT on farmers' organic conversion decisions, although the research findings regarding this are scarce to our knowledge.

The ICT supports all stakeholders of agri-food value chains to contribute to effectively achieving the sustainability objectives, because they have integrated, unmediated access to vast amounts of large-scale data sets of diverse types from a variety of sources that enable them to generate value and extract insights from these data [80]. The Internet of Things, blockchain, big data, and cloud computing technologies are driving the agricultural supply chain towards a digital supply chain environment that is data-driven [81]. For this reason, digitalization has become a crucial issue for food retailers, distributors, processors, and farmers, as it enables them to compete more effectively by improving the economic efficiency, transparency and traceability, environmental and social performance, legal culpability, and e-market/supply accessibility [82].

The rapid progress in ICT provides sets of applications in all of the steps of the food supplying process, from farm businesses to retailers. Thus, digitalization allows the food supply chains to be highly connected, efficient, and responsive to customer needs as well as to regulatory requirements. For example, food retailers are supported in their sourcing strategy, because it is easy to keep track of a supplier's performance based on various parameters such as cost, quality, availability, innovativeness, and environmental impact [78]. Food franchises largely have the advantage of shared resources of their suppliers in managing orders, payments, inventory, and after-sales services [83]. Food producers are supported to manage orders and real-time decisions related to shipments and to have access to the point of sale, while the use of big data processing enables them to make more informed production decisions [84].

At the farm business level, there exist sets of applications in all kinds of enterprises, operations, and activities. The combination of the Internet of Things and blockchain leads to more autonomy and intelligence in managing precision agriculture in more efficient and

optimized ways [85], while the supporting or monitoring systems that are created help farmers to more effectively and efficiently carry out fertilization and irrigation, protection of crops from pests, monitoring of livestock using wireless sensor networks, implementation of certification systems, and management of their businesses. The scanning of codes printed on food packaging and linking the barcode to a food category or a certain batch is also a useful method for farmers, marketers, and consumers, while several application areas by means of mobile appliances, such as extension services, precision agriculture, e-commerce, and information services, facilitate food production, marketing, and bargaining. New devices, such as tablets, and new services, such as cloud computing, augmented reality, and near-field communication, have great potential in agriculture and food production [30,39,86–89].

Taking into account all of these aspects in combination with the importance of technology for organic conversion decisions [33] and the positive impact of the use of computer technologies in farmers' eco-certification decisions [46], it is expected that the availability of many types of ICT applications will positively affect organic conversion decisions at all steps and activities of the production and supplying processes.

### 4.2. Institutional Factors

Some institutional factors, such as the rules (formal and informal) that structure social interactions, as well as laws, social norms, customs, and rules of thumb, can affect farmers' conversion decisions [47,62–64]. The institutionalization of organic agriculture over that last three decades has mostly occurred through the development of the market for organic products, the creation of markets for standards, certifications, and accreditation systems. A hybrid governance structure can be seen whereby actors with conflicting interests, visions, and projects compete in the field but actually tend to converge. Despite the conflicts between public and private actors over the control of activities, they are engaged in the active construction of markets and the facilitation of their expansion. This is part of the institutionalization of the organic sector, and it has some important performative effects [48].

Some research reveals more precise findings in regard to the interaction between certification and the norms of production and marketing. A necessary condition for increasing smallholder participation in markets is the national institutions to support the compliance of farmers with certification standards that reflect a market demand [30]. Trust and reciprocal relationships through high-quality communication and well-functioning institutional arrangements play pivotal roles for partnerships to be maintained [49]. The interaction of standards with pre-existing norms of production and trade enables us to understand how the social norms exert more influence in uncertain decisions such as farmers' organic conversion decisions [30]. For example, pro-organic social norms and attitudes, such as the attitudes of household members and other farmers towards organic farming, substantially impact on farmers' conversion decisions [45,63,64]. Of particular note are the special importance of organized organic markets [24], the EU institutional framework for organic conversion, and the role of farmers' perceptions in regard to institutional support [33,61].

### 4.3. Social Networks and Knowledge Transfer

Supportive social networks motivate the organic transformation of agriculture [26]. It has been shown that societal networks in developed markets, such as the Dutch dairy market, have been particularly influential in advancing organic food supply. Such networks are almost absent in less developed countries, such as Thailand [53], although their impact on farmers' choices to adopt organic practices is decisive [33]. Frequent communication among farmers in a network is linked with the higher importance of organic farming information received from formal actors than to information received from informal actors [54].

More specific findings are those of Sutherland and Darnhofer [27], Tzouramani et al. [20], and Nalubwama et al. [24]. They suggest that the fast expansion of organic

farming in some areas is partially attributed to the development of agricultural information provision and advisory systems and to the creation of rural networks that facilitate knowledge creation and transfer. This was also confirmed by Luh et al. [61] and Duvaleix et al. [52], indicating the high importance of knowledge transfer and information provision for farmers' decisions related to organic farming and environmental friendliness. In contrast, the low density of organic farms in an area combined with a lack of peer networks to provide support is a major barrier to organic conversion [32]. Of special importance for organic conversion is the finding that farmers make technical changes that affect their productions by imitating other credible farmers [63].

### 4.4. Relationships between Market Players and Institutions

Given the high importance of relationships and collaboration among food supply chain stakeholders [76], we chose to introduce relationships in the framework (Figure 1) as a separate set of external factors. The interactions between consumers' preferences, trade flows, and supply chain structure enable us to understand why the farmers of a certain country and/or industry, such as Brazilian soybean producers, adopted environmental certification programs on a larger scale than other nearby countries [40]. More specific findings regarding individual farmers' behaviors reveal that the farmers' decisions related to organic conversion reflect the interaction of perceptions, relationships, policies, and economic factors, which either enable conversion or provide barriers [38].

As farmers are continuously facing global market conditions and tightening regulations, the collaborative partnerships that contribute to adaptability and flexibility become more essential. For example, the interactions between food producers and certifiers [31] substantially impact organic conversion decisions, while various forms of cooperation enable farmers to counteract the risks caused by the growing presence of large wholesalers, intermediaries, and retailers in the market that could pose a threat to organic farms [49,76]. These partnerships, as previously noted, are facilitated and maintained if a climate with trust and reciprocity is coupled with high-quality communication and well-functioning institutions [49]. Of particular importance for organic conversion decisions is collaboration between farmers, because certain benefits can arise from it [34,36,67,90], making the organic conversion more attractive to farmers. For instance, organic certification becomes more likely in cases where the certification process is organized by a farmer's organization that allow the less educated farmers to participate in the standard adoption, thereby reducing the influence of education [37].

### 4.5. Financial Factors

In the present set, selected financial factors were included, because the economic capital linked with some financial aspects of the external business environment were found to be important for farm business owners when considering the conversion of conventional farming to organic farming [22,71]. For example, the rapid development of intensive primary production, encouraged by financing options that the local banks provide, results in high land prices, which negatively affect farmers' organic conversion decisions [35]. The impact of economic pressures on conventional farming causes farmers to reduce, rather than intensify, the input use. This reflects a change in norms towards balancing the risks and potential returns rather than optimizing production [55]. Thus, in some cases, the high cost of inputs in conventional farming can positively affect the decisions of farmers to adopt organic practices [31,62], while the financial risk during the conversion period is a barrier to adoption decisions [26,36].

In addition to the above, the scarcity and high cost of organic inputs [34] as well as the high price of local land linked with scarcity negatively impact conversion decisions [35]. Taking into consideration the differences that exist between regions and countries in regard to economic conditions [1,2,15,16], the financial system can encourage or discourage farmers to proceed with organic conversion. For example, in some less developed countries,

the absence of an adequate mechanism for farmers to access finance services is a factor that negatively impacts conversion decisions [27,42].

*4.6. Public Policy*

Following the structure of the framework (Figure 1), the public authorities schedule changes in relation to selected factors of the farm business environment that impact on conversion decisions. Thus, they build a policy mix that can include regulations, standards (such as eco-brands, e.g., biolabels), direct subsidies to producers, input taxes, research funding, training and information provision, financial support for investments, and sponsorship of communication instruments (such as promotional campaigns and consulting) [24,28,34–36,50–52,55,68,69]. Given that shifts in agricultural policy cause habit changes (directly or indirectly) and that certain policy regulations motivate higher organic adoption rates [27], the present set focused on factors found to directly affect conversion decisions.

Subsidies are powerful horizontal instruments that motivate the adoption of organic techniques and conversion decisions [24,28,50]. For example, in an earlier published study, it was found that an increase in subsidies motivates adoption in poorer Spanish farms. If the farms were to receive EU-average subsidy levels, this could motivate adoption in a substantial number of farms. Of interest is the finding that early adopters correspond to wealthy farmers. However, as economic conditions for conversion improve, poorer farms also shift to organic activities [28]. This implies that the provision of horizontal subsidies, without taking into consideration the heterogeneity among farm businesses, is not expected to be sufficiently effective and efficient.

The financial compensation required to reward farmers for the environmental performance, cost increase, and decreases in yields can impact farmers' decisions, positively or negatively. This depends on the uncertainty in producers' expectations about future returns and about the impact of policy changes on these expectations [24]. Thus, public policy can be a source of uncertainty in some cases, negatively impacting farmers' decisions [24]. For instance, when the future of a policy program is uncertain, its introduction increases the value of waiting to convert. This lowers the conversion rates [51]. The continued existence and increase in the number of environmental certification programs offer strategic value over the long term, enhancing the producer's reputation. Thus, the organic production and marketing shares increase. Of particular importance are the well-designed sectors or product-based programs, such as the EU biolabel and PDO (protected designation of origin) and PGI (protected geographic indications) labels that add value for primary producers [51,52]. Such programs facilitate farmers to use local, regional, and national brands for their organic products thus strengthening their position in the marketplace.

## 5. Internal Farm Business Environment

The internal farm business environment is composed of various elements present inside the farm business organization, which are factors that can affect farmers' choices, activities, and decisions and can be affected by them. These factors include the value system, vision, mission and objectives, organizational structure, culture, human resources, physical resources, and technological capabilities. Zagaria et al. [76] defined them as farmers' assets, such as age, farmers' goals, values and attitudes, farm and field characteristics, and perceived adaptation options. We focused on the factors that were found to affect farmers' conversion decisions. Based on many studies exploring them in various agricultural industries, locations, regions, and countries [24,26–33,35–37,45,47,50,51,56–62,64–67,70], we identified sets of factors that determine the decisions of farmers to convert conventional farming activities to organic activities worldwide.

The factors were grouped into those related to the farm business characteristics and those related to the characteristics of the farmers. This enables public policies and private strategies to be properly customized in terms of farm business segments based on the characteristics that are the most appropriate and relevant to the case; thus, acceleration can

more effectively and efficiently be achieved. We borrowed ideas from models that connect farmers' decisions with characteristics of the farm businesses [36,47,59] in combination with the studies mostly focusing on policy issues [17,31,55,68] to introduce these factors into the framework we built (Figure 1, Table A1). The related definitions are presented below.

### 5.1. Farm Business Characteristics

Some of the farms' characteristics were objectively measured and easily recorded, while some others needed special research to be identified. The farm business characteristic that was found in most studies to affect organic conversion decisions was farm size, measured in physical and/or economic units. More specifically, smaller farms were more willing to proceed with conversion [29,30,59,64], while farmers with a large or medium farm level of production did not change their strategy to proceed with organic conversion [31,65,70]. Depending on the circumstances, farm size may have a positive impact on conversion such as in countries in which the mean farm size is too small [66]. The type of enterprise was also a determinant of conversion decisions in most studies [37,56,57,60,64,70], but it was not reported as a determinant [47] in some other studies. The number of plots, enterprise productivity, production, certification costs, and profitability as well as the income per product quantity and the low yields of organic production have been reported as determinants of farmers' conversion decisions [26,29,31,33,36,37,45,50,57,58,64].

The use of technologies, including information and communication technology; access to technological support, credit, and extension services; the location of the farm business activity determined conversion decisions [30,31,33,36,37,60,70]. In regard to the location, farm businesses in less favored areas as well as those that were a short distance to an urban center or a market [56] tended to be more willing to switch to organic production, while farms that were situated in a high population density area could be less willing [35,59]. The type of farming was also found to determine conversion decisions. More precisely, the intensity of farming, specialization, and the farms' strategy to diversify the production were determinants of the conversion decisions of farmers [27,29,31,57], while the increased need for human labor in organic farming was reported as a barrier [26]. The fact that farmers that own their land are more likely to invest in long-term measures, such as organic certification [37], led to its inclusion in the characteristics that affect farms' organic conversion.

### 5.2. Farmers' Characteristics

Several sets of farmers' characteristics were identified to affect conversion decisions worldwide. These can be distinguished into demographic and other social characteristics including professional skills and psychographic and behavioral characteristics. The former can be more objectively measured and easily identified and recorded than the latter, thus allowing the farmers to be easily categorized in a way that clearly distinguishes each segment from the other.

### 5.2.1. Demographic and Other Social Characteristics

The sets of farmers' demographic and other social characteristics, including professional skills, that have been explored in many studies, were found to impact on farmers' decisions regarding proceeding with organic conversion. More specifically, farmers' age and experience (which can be measured in years of farming), household wealth, and household size (which is linked with the available workforce), as well as gender and off-farm activities, were important determinants of farmers' decisions [26,27,30,47,57,58,60,62]. Young farmers, a higher level of education and/or special agricultural training, and the use of ICT were linked with a higher intention to proceed with organic conversion [30,47,56,58,59], while the household labor limitation constrains such adoption [62]. In regard to gender, it was revealed that, in some cases, a higher percentage of women than men proceed with organic conversion [58].

Because knowledge and information substantially contribute to replacing synthetic agrochemicals in organic farming, the limited awareness and knowledge of organic farming can constrain its adoption [28,62]. Thus, knowledge, education, and training of farmers were included among the most important factors that determine conversion decisions. Apart from farmers' education and training, the development of their knowledge comes largely from what they learn through their experiences and via social learning [31,47,51,56–61]. Notably, in a developed country, such as the US, each additional year of farming or year of education of fruit and vegetable producers leads production under organic certification to increase by 0.33% and 1.15%, respectively [31]. In contrast, in some cases, such as in Vietnam and Nepal, the level of education of tea producers negatively affects organic conversion decisions [31,66], but their participation in special training programs positively affects them [31].

The relationships of farmers, exemplified by their participation in community organizations and the benefits they receive from membership, also determine organic conversion decisions [30,47]. The same stands for their relationships and interaction with social agents, which are important for their conversion decisions. More specifically, farmers' attitudes regarding environmental practices, which evolve over time, are influenced by their previous individual experiences and observations of other farmers' environmental practices [27]. Some other variables, such as the use of the Internet [30], farmers' livelihood assets, and their vulnerability contexts in combination with livelihood activities and gender-related parameters simultaneously shaped the adopted decision-making process [62].

### 5.2.2. Psychographic and Behavioral Characteristics

The psychographic and behavioral aspects that determine farmers' conversion decisions are related to farmers' attitudes and perceptions, values, beliefs and motives, orientations and strategies, relationships, and cooperation. Farmers' attitudes regarding organic farming, linked with the recognition of the benefits of organic production, were a major determinant of conversion decisions [36,47] as well as their attitudes regarding information, the environment, and health. The more active the farmers were in regard to acquiring information about the economic viability of organic farming and the more highly they regard the environment and health, the more likely they were to adopt organic farming [56,61,62].

The pro-organic ideology of farmers and their beliefs regarding the philosophy of the organic, their willingness to preserve the environment and generate local employment, and their satisfaction with the incentives for organic farming and sale prices positively impacted on conversion decisions [47]. It was found that farmers that strongly believed in the philosophy of the organic had, on average, 13% more production under certification [31], while the likelihood that farmers will carry out organic rather than conventional production was almost 2.4 times greater if the farmers' goal was "sustainable and environmentally friendly farming" [56]. These farmers were willing to risk some of their income and adapt crop and animal management practices, as necessary, to overcome a variety of challenges [47].

Depending on regional circumstances, farmers adopt organic practices on the basis of economic motives, such as in the case of US (Virginia) farmers, who are most likely (69%) to adopt organic techniques if the perceived benefits are high and the costs are low [45]. The dissatisfaction of farmers with the present conventional farming, the belief that the technical problems they face are sufficiently solved, and a range of social factors (such as the favorable attitudes regarding organic farming of household members and other farmers) positively affect the organic conversion choice [32].

Similar findings are provided by some studies mostly focusing on farmers' perceptions. For instance, farmers' perceptions that the utility of organic production is higher than conventional production as well as the perception of the risk of conventional farming and the harmful effects of pesticides on food quality have a positive influence on organic conversion [36,56]. The farmers' economic concerns of organic farming, including their

perception of the economic incentives of expecting higher return, play a crucial role in the development of organic practices [33,60,61]. Important factors that affect the adoption of organic farming are also farmers' perceptions of the characteristics of technology, the institutional support for socio-technical learning networks, and the credit gained by the non-governmental organizations that promote organic farming [33]. Because farmers' groups and cooperatives help them to efficiently sell their products as well as to mitigate some market-related threats and weaknesses, which are due to certification costs and required knowledge, the horizontal and vertical cooperation can be a factor that determines the conversion decisions [36,66,67].

Factors that are found to negatively impact on organic conversion have also been reported in some studies. Farmers' ideology regarding organic farming, the financial risks (especially during the conversion period), the skeptical attitudes regarding social networks, the uncertainty of the environmental benefits of organic production, and the institutional factors and communications from regulatory institutions are of particular importance [26]. The sources of perceived risks on the farms, such as erratic rainfall, limited knowledge, and market for organic products [24] as well as the disincentive of expecting higher costs play a crucial role in adoption decisions [61].

A major barrier to organic adoption is farmers' perception that friends, families, and other farmers are not supportive of organic farming [32,61] due to the fact of its economic impacts and lack of stability in farm income [61]. Barriers related to organic certification are the belief that the organic certification process is confusing [31], time-consuming, and expensive [45]; animal feed is hard to obtain; the benefits are not worthwhile; and the certification has little meaning [36]. Indicative is the result that producers who perceive that costs and interaction with the certifier are barriers to entry for organic markets have approximately 6% less production under certification [31]. In regard to the goals of farmers, the probability that they will conduct organic production is reduced if they aim to "achieve a reliable and stable income", "maximize profit", or "improve the farm for the next generation" [56]. They stay with their current strategy if they do not consider organic production as a substantially better strategy [65].

## 6. Discussion

The determinants of farmers' organic conversion decisions categorized into factors of the external and internal farm business environment and further organized into sets and sub-sets were integrated in the farmers' decision process, as shown in the framework presented in Figure 1 and Table A1. The framework we built distinguishes itself from past work that explored farmers' organic conversion decisions, primarily through the integration of the numerous sets of factors that determine conversion decisions worldwide. Following Zagaria's et al. [76], it provides a conceptual structure and a causal foundation to understand the farmers' organic conversion decisions that are based on their perception of changes to the external farm environment and are implemented with repercussions to internal and external farm environment characteristics. Thus, the present study continues the academic discussion [87–89] that focuses on the combination of the theory of farmers' behavior with the agency theory to connect farmers' adoption decisions related to environmental issues with agricultural systems as complex social–ecological systems.

Because each of the previous studies focused on a limited number of farmers' behavior determinants, while farmers' organic conversion decisions are influenced by numerous sets of physical, economic, social, and psychological factors of external and internal farm environments, it cannot be argued that some important factors have not been omitted in each case previously studied. This does not allow sufficient information to public authorities to be provided, and subsequently, the highest effectiveness and efficiency may not be achieved in policy implementation. The factors identified determine the potential of farm businesses to integrate the ecological, social, and economic outcomes of organic farming into farmers' conversion decisions [4,5,7–13,16,19,76,91–95], while some of them

go beyond those that have been taken into account by the Common Agricultural Policy for organic farming [2,24,27,28,34–36,50–52,55,58,68,69,96–101].

The group of external factors includes those that the farmers cannot control in the short term, but they properly respond to their changes. By contrast, they can manage the factors of the internal farm business environment.

### 6.1. External Farm Business Environment

The factors of the external business environment are related to the market such as the demand, the distance to the market or point of selling, the price of organic food, the supply chain, certification systems, and technologies. This group also includes sets of factors that are related to the institutional arrangements, social networks, information provision and advisory systems, relationships between market participants and institutions, financial environment, and public policy. Public policy seeks to bring about changes in these factors and, thus, indirectly influences farmers' decisions that attempt to properly respond to external changes. It also directly impacts conversion decisions via certain policy measures that include regulations, standards, subsidies to producers, taxes on inputs, financial support for investments, research funding, training and information provision, compensation, and supporting programs and projects [1,2,24,28,34–36,50–52,55,58,68,69,96].

Some indicative strategies, policy measures, and actions that can, directly or indirectly, cause changes in the external determinants of farmers' conversion decisions were formulated by borrowing ideas from previous studies [2,24,27,28,34–36,50–52,55,68,69,98–101]. These are epigrammatically presented on Table 3, in accordance with the sets of factors of the external farm environment that impact on farmers' decisions.

**Table 3.** Strategies, policy measures, and actions that can cause changes in factors of the external farm environment.

| Determinants | Strategies, Measures, and Actions |
| --- | --- |
| Market-related factors | Organic market development—local and farmers' markets<br>Organic consumption promotion campaigns<br>Direct subsidies to organic producers<br>Financial support for cost reduction and competitiveness<br>Development of organic product supply chains<br>Reduction of certification-related bureaucracy and costs<br>Eco-friendly technology development and diffusion, especially ICT<br>Eco-friendly brand development, based on organic character<br>Research funding for organic production, supply, and consumption<br>Organic input market development<br>Education and training<br>Information provision<br>Facilitating the development of farmers' organizations |
| Institutional factors | Development of institutional arrangements and services that:<br>Support compliance with organic certification requirements;<br>Encourage eco-friendly innovation development;<br>Enhance organic food traceability and market transparency;<br>Facilitate communication among food supply stakeholders;<br>Promote collaboration and trust among organic food supply stakeholders.<br>Institutions related to research and education<br>Performance and efficiency of organic input markets<br>Institutional arrangements for protection of natural resources and environment |

**Table 3.** *Cont.*

| Determinants | Strategies, Measures, and Actions |
|---|---|
| Social networks and knowledge transfer | Facilitating and supporting: <br> Network development and operation; <br> Knowledge development, exchange, and diffusion; <br> Increase density of organic farming in selected regions. |
| Relationships between market players and institutions | Promoting and facilitating: <br> Encouragement and support for collaborative partnerships; <br> Relationships between farmers; between farmers and marketers; between farmers, marketers, and certifiers, research, educational and information provision institutions. |
| Financial factors | Efficient financial markets <br> Facilitating farmers to have access to efficient financial services <br> Support for financial provision to farmers and other food supply chain participants |
| Policy-related factors | Direct subsidies <br> Lowering uncertainties in producers and marketers' expectations <br> Biolabeling and certification programs <br> Directions and regulations for protection of natural resources and the environment |

The literature reveals that the importance of factors that impact farmers' organic conversion decisions substantially differs between farm businesses, locations, regions, and countries. The most important differences in external factors concern access in an organic market with a high demand [20–23,31,82] in combination with the distance to the market or point of sale [23,24,29,30,32,35] and the supply chain [39,40]. This, combined with well-functioning institutional arrangements, such as rules, certification bodies, research and education organizations, and supportive services and networks, creates an external farm business environment conducive to the spread of organic farming, such as in the case of Denmark and Switzerland [16,25,26,33,38,41,49]. Farms far from markets with high organic food demand and facing a lack of well-functioning institutions, supportive organizations, services, and peer networks operate in conditions less favorable for organic conversion.

*6.2. Internal Farm Environment*

The factors of the internal environment are categorized as farm businesses' characteristics and farmers' characteristics. The former includes some physical and economic characteristics of farms, such as the location, size, productivity, intensification, and specialization. The latter include social characteristics, such as age, experience, education, knowledge, professional skills, off-farm activities, household size, relationships, and co-operation. Some important psychographic and behavioral characteristics of farmers are the profit orientation, attitudes regarding organic farming, environmental attitude, risk attitude, information seeking attitude, beliefs regarding the philosophy of the organic and pro-organic ideology, perception of social agents, attitudes of the household members and other peers, willingness to preserve the environment and generate local employment, and satisfaction with the incentives for organic farming and product sale prices. Changes and improvements in these factors can be brought about directly by the farmers themselves, while they can also be triggered, facilitated, supported, and encouraged by public authorities, market participants, and other social agents.

Some indicative strategies, policy measures, and actions that can, directly or indirectly, cause changes in the internal determinants of farmers' conversion decisions were formulated by borrowing ideas from previous studies [2,24,27,28,34–36,50–52,55,68,69,98–101]. These are epigrammatically presented on Table 4, in accordance with the sets of factors of the internal farm business environment that affect farmers' decisions.

**Table 4.** Strategies, policy measures, and actions that can cause changes in factors of the internal farm environment.

| Determinants | Strategies, Measures, and Actions |
| --- | --- |
| Farm business characteristics | Segment farm businesses according to the geographic place, size, and enterprise<br>Support and facilitate changes to be caused in selected farm business characteristics |
| Farmers' demographic and social characteristics | Segment farmers according to the age, experience, income, household size, gender, off-farm activities, education and training, and ICT use<br>Support and facilitate changes to be caused in selected farmers' characteristics<br>Support the process learning by experiences and social learning, observation of other farmers, and interaction with social agents<br>Focus on segments more willing to proceed in organic conversion to achieve a high conversion rate in the short run<br>Focus on segments less or not willing to proceed in organic conversion to increase effectiveness in the long run |
| Farmers' psychographic and behavioral characteristics | Segment farmers according to regional circumstances<br>Support to education, training, and information provision<br>Support for research and knowledge development<br>Support actions facilitating changes in attitudes toward organic, health, and environment; increase awareness with organic by focusing on economic, environmental, and social outcomes of organic farming |

Because the policy measures aim to mostly motivate farmers with subsidies, to compensate them for income losses and to reward them for beneficial services provided to ecosystems and societies [2,24,28,50,51], it is worthwhile to note that farmers do not adopt organic practices for solely economic reasons. They proceed with organic farming if they perceive economic benefits to be high and costs to be low [45] and, in some cases, as well as if they are dissatisfied with the present conventional farming practices [32]. In contrast, in some other cases, pro-organic ideology [26] and the farmers' goal of "sustainable and environment-friendly farming" [56] were the most important determinants of conversion decisions.

*6.3. Dynamics and Interactions*

The aspects of the external farm business environment as well as the farms' and farmers' characteristics evolve over time. The mean age of farmers and their education and knowledge can change, the regional or national systems providing training and information can be improved, the available technologies can be rapidly multiplied, and the consumers' income and interest in organic products may change. Country-level and global shifts in consumption, marketing, production, and trade lead to organizational changes along food chains [19], while increases in efficiency in supply chains lead to the ability to deliver high volumes of products at low prices worldwide [78]. The changes in habits linked to wider societal trends, with shifts in agricultural policy [27], as well as the changes in economic conditions that favor conversion can attract the interest of new farms to shift to organic activities [28]. This implies that the importance of the determinants of conversion decisions can change over time. Thus, new research challenges arise related to the integration of the time-related changes [74,76] into the farmers' decision framework, highlighting its dynamic character.

The identified factors interact with each other, enabling synergistic or antagonistic results to be revealed [58,63,64]. For instance, the interaction regarding the utility difference between organic and conventional farming that farmers perceive and their environmental concern does not enable the environmental concern to affect the probability of organic adoption if this difference is positive [64]. Most examples concern interaction between

internal and external factors. Major interacting barriers in an area with a low density of organic farms are the lack of supply and delivery points within an acceptable travel distance and the lack of peer networks that provide informal support [58], while the price of the land related to its local scarcity interacts with the distance of the farm as well as cooperation and exchange of experience among organic farmers [35]. The technical abilities of farmers and the capacity of agricultural advisers to cope with psychological factors associated with the institutional support systems motivate farmers to adopt organic-related innovations [33]. The review reveals that although the interactions were of particular importance for farmers' decisions, in only a few cases have they been studied.

## 7. Conclusions

The present study helped to answer the question of how the spread of organic farming could be accelerated by identifying the determinants of farmers' organic conversion decisions. We integrated the numerous factors that affect farmers' behavior related to organic conversion worldwide into a framework that provides a theoretical foundation with a more holistic approach to be adopted in policy scheduling and scientific research. The framework, which includes sets of factors that were identified to impact on farmers' decisions to convert to organic farming activities, provides comprehensive information that helps public authorities choose among several sets of factors to which they will bring about changes resulting in the acceleration of organic conversion decisions.

This framework also enables policy effectiveness and efficiency to be holistically assessed. For example, a policy evaluation that is based on physical and economic criteria, such as the changes in organic farmland, livestock, and consumption shares, does not provide sufficient information if it does not take into account the changes in psychological aspects such as the farmers' beliefs, attitudes, and perceptions towards organic farming. Because each of the previous studies exploring the determinants of farmers' organic conversion decisions is focused on a limited number of factors in a narrow geographic area, which implies that some important determinants may have been omitted in some cases, the present article contributes to filling this gap that resulted from the insufficiency of information in previous studies.

The farmers' decision framework we built can facilitate researchers to select from a wide range of factors those which are most likely to influence the conversion decisions of the farmers they aim to study. In this way, the research findings can provide sufficient information to public authorities supporting them to achieve the highest possible effectiveness and efficiency in policy implementation. Apart from public policy that is related to the organic farming, the framework can also be helpful in policy planning and assessment that focuses on other types of ecological farming, with or without certification, such as farming that is low-input, conservative, integrated, agroecological or biodynamic [96,97,99–101]. This can also facilitate academics that aim to explore farmers' and small food firms' decisions in regard to the implementation of some other environmentally friendly systems such as integrative farm assurance, integrative farming, sustainable farming, ISO 14001. The framework can also support farmers' cooperatives, food producers, and marketers, who supply food in a totally globalized business environment, to efficiently implement eco-friendly strategies by effectively encouraging farmers, being their suppliers, to participate in organic conversion or in other eco-friendly farming schemes.

The sets and sub-sets of factors that impact farmers' decisions related to the conversion of conventional farming activities to organic activities incorporated in the farm business decision framework are categorized as those located at the external farm business environment and those located at the internal environment. From the numerous factors related to the external environment, the most important are the demand and price of organic products, the distance of farming activities to the market or point of selling and, broadly speaking, access to markets, the available technologies, education provision in combination with knowledge transfer, peer networks, and society's attitudes regarding organic farming

as well as the provision of subsidies and farmers' compensation for the higher cost and the lower yields.

A number of aspects of internal farms' business environments—some of them can be easily recognized—enable public authorities, private actors, and social actors to properly customize their efforts as well as to choose the target segments in order to accelerate conversion decisions more effectively. The most important characteristics of farms and farmers include farm location, farm size and enterprise, expected costs and profits, knowledge and ICT use, farmers' age, education and gender, off-farm activities, attitudes regarding organic farming, willingness to preserve the environment, beliefs regarding the philosophy of the organic, and attitudes of household members and other peers. Of particular importance is farmers' satisfaction with the incentives for organic farming and product sale prices, as well as their perception in regard to technical problems and the certification process.

Given the large number of factors that determine farmers' conversion decisions, public authorities, market players, and social actors aiming to promote organic farming should take into account the fact that the highest effectiveness and efficiency cannot be achieved if their efforts are restricted to a narrow range of factors. Instead, each effort should include a combination of strategies, measures, and actions that correspond to the most important of the factors that influence farmers' conversion decisions in each case. Because the importance of the determinants of farmers' decisions substantially differs among farm businesses, enterprises/industries, locations, regions, and countries, the policies, strategies, and actions for acceleration cannot be sufficiently effective if these are not properly customized to the specific case. More precisely, these should be scheduled to change selected factors that have the highest impact on farmers' conversion decisions, simultaneously focusing on targeted farm business segments. Thus, properly designed research findings are needed in each case, following a more holistic approach in the exploration.

It is of particular importance to note that factors influencing conversion decisions interact with each other, thus revealing synergistic or antagonistic results, while the aspects of the external farm business environment as well as the characteristics of the farm evolve over time. Thus, public authorities, market players, and social actors must not neglect such interactions as well as time-dependent changes. The interactions and time-dependent changes also constitute important challenges for future research including also a dynamic extension of the proposed framework. In addition to the above, some factors that were not found to affect organic conversion decisions were found to affect farmers' decisions to implement integrated management systems [4,46,67,94,102], which are eco-friendly systems. This could be attributed to certain difficulties that organic production and the certification process face [31,43,44,67]. Thus, further exploration of organic production and certification practices that farmers perceive to be difficult to adopt could reveal additional decision determinants. Given the rapid evolution of ICT and global threats, such as climate change and pandemics [103,104], additional determinants of organic conversion decisions may be revealed, offering new research opportunities.

**Author Contributions:** Conceptualization, P.K.; methodology, P.K.; software, S.K.; validation, P.K., S.K.; formal analysis, P.K.; investigation, P.K., S.K.; resources, P.K.; data curation, P.K.; writing—original draft preparation, P.K.; writing—review and editing, P.K.; visualization, S.K.; supervision, P.K.; project administration, P.K. Both authors have read and agreed to the published version of the manuscript.

**Funding:** This research received no external funding.

**Institutional Review Board Statement:** Not applicable.

**Informed Consent Statement:** Not applicable.

**Data Availability Statement:** Not applicable.

**Acknowledgments:** The authors thank the four anonymous reviewers and editor for their helpful comments.

**Conflicts of Interest:** The authors declare no conflict of interest.

**Appendix A**

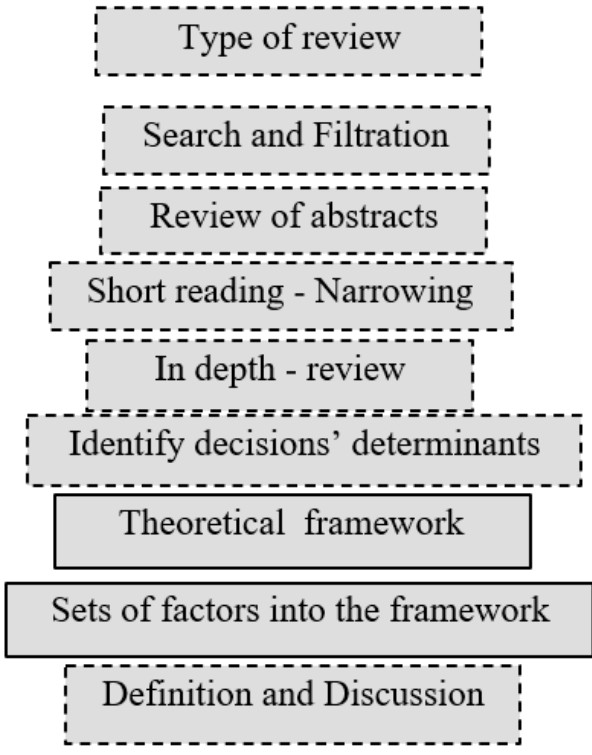

**Figure A1.** The steps of the study.

**Table A1.** The sets of factors that affect farmers' decisions to convert conventional farming activities into organic activities.

|  | Sets/Sub-Sets of Factors | References |
|---|---|---|
| **External factors** | Market factors: |  |
|  | *Demand* | [20–22,31,34,35,37,64] |
|  | *Price* | [23,25–28,37,70] |
|  | *Distance to market or point of sale* | [23,24,29–32,34,35,37] |
|  | *Supply chain* | [38–40] |
|  | *Certification schemes* | [22,26,31,37,38,41–44] |
|  | *Technologies* | [30,33,36,46] |
|  | Institutional factors | [24,30,33,45,48,49,61,63,64] |
|  | Social networks and knowledge transfer | [20,24,26,27,32,33,52–54,61,63] |
|  | Relationships between market players and institutions | [31,34,36–38,40,49,67,96] |
|  | Financial factors | [22,26,31,34–36,62] |
|  | Public policy | [24,27,28,34–36,50–52,55,68,69] |
| **Internal factors** | Farm business characteristics | [26,27,29–31,33,35–37,45,47,50,56–60,64,65,70] |
|  | Farmers' characteristics: |  |
|  | *Demographic and other social characteristics* | [26–28,30,31,47,51,56–62,66] |
|  | *Psychographic and behavioral characteristics* | [24,26,31–33,36,45,47,56,60–62,65–67] |

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
