# Peer review of "Factors that Impact Farmers’ Organic Conversion Decisions"

_sustainability, doi:10.3390/su13094715_

Round 1
Reviewer 1 Report
- I think the paper has improved it its revised version as the authors claimed. A few comments are the following:
- The information contained in Figure 1 can be presented in text with much concise form and as such the figure does not add much. At best, the figure can be included in Appendix.
- Similarly, I would suggest including Table 1, 2 and 3 in Appendix.
- Table 4 and 5 may remain in the text.
- With sufficient changes and the development of the paper in its revised version, I believe, it has become rich enough that readers will benefit.
Author Response
Point 1: think the paper has improved it its revised version as the authors claimed. A few comments are the following:
Response1: Thank you very much because with your comments you helped us to improve our manuscript. We have further elaborated our manuscript in line with what you have suggested.
Point 2: The information contained in Figure 1 can be presented in text with much concise form and as such the figure does not add much. At best, the figure can be included in Appendix.
Response 2: Thank the reviewer for the valuable comment and suggestion. The steps the Figure 1 showed has been described in the text (section 2, especially lines 141-150) while the figure is moved in Appendix A (lines 172-173 & lines 833-848).
Point 3: Similarly, I would suggest including Table 1, 2 and 3 in Appendix.
Table 4 and 5 may remain in the text.
Response 3: Thank the reviewer for the valuable suggestions.
We have moved the table 3 in Appendix A (lines 849-851), but we keep tables 1 and 2 in the text, because these include arithmetic data that lead important conclusion to be revealed (lines 646 & 762).
We keep table 4 and 5 (renumbered) in the main body of the manuscript, as you have suggested.
Point 4: With sufficient changes and the development of the paper in its revised version, I believe, it has become rich enough that readers will benefit.
Response 4: Thank you very much!
Reviewer 2 Report
This is my second review of this article. During the first submission many issues have been detected and most have been corrected by the authors.
At the current stage the manuscript has been clearly further supplemented and expanded. Key modifications concern such sections as the methodology of the research and discussion. Overall, in my opinion, it is a valuable contribution as it aggregates numerous studies, while also being a synthesis conducted by the authors. Therefore the article could be recommended for publishing at this point, but due to modifications several things need to be corrected or looked at before the publishing.
First, I do not fully agree with the first sentence of the conclusions, namely “The present study answered the question of how the spread of organic farming could be accelerated, by focusing on the determinants of farmers’ organic conversion decisions”. The article doesn’t answer this question, it only reviews the factors that influence farmers’ decisions, but it doesn’t evaluate the significance of each factor. The article provides a review and synthesis of possible factors.
Second, while it seems too late for the research process, I believe the applied approach and key words used to search for articles concerning the farmers’ decisions towards transition to organic agriculture was not selected in the best possible way. Word combinations could have been selected in a way to generate more search results, if only the authors have chosen more options. In this view the outputs of the Horizon 2020 LANDSUPPORT project (https://www.landsupport.eu) could be of use. Also, limiting conversion to only “organic” has narrowed it further, while close ideas of ecological farming covering low-input, conservative, integrated, agroecological have been omitted.
Third, I would still suggest to look at the Horizon 2020 LIFT project’s results (https://www.lift-h2020.eu), as it is one of the key contributions on the subject at the moment and I’m honestly confused why the authors haven’t looked at it. It deals precisely with the decisions of the agricultural transformation towards ecological approaches and has been publishing outputs since 2018. The second project the outputs of which are omitted here is the Horizon 2020 UNISECO project (https://uniseco-project.eu), which also had made valuable contributions to agroecological farming theory and practice, gathering numerous outputs from stakeholders in regard to transition process. Both project have published reports and scientific articles.
Fourth, I am not sure the Figure 1 is the best way to depict the method applied.
Fifth, there is an issue with Figure XX, as words at the bottom are only partially visible. Same with Table 1, here the issue is with the text on the left side.
Sixth, a professional English proof-reading is needed, as the style and grammar are not polished yet.
Author Response
Point 1: This is my second review of this article. During the first submission many issues have been detected and most have been corrected by the authors.
At the current stage the manuscript has been clearly further supplemented and expanded. Key modifications concern such sections as the methodology of the research and discussion. Overall, in my opinion, it is a valuable contribution as it aggregates numerous studies, while also being a synthesis conducted by the authors. Therefore the article could be recommended for publishing at this point, but due to modifications several things need to be corrected or looked at before the publishing.
Response 1: Thank you very much because with your comments you helped us to substantially improve our manuscript. We have further elaborated our manuscript in line with what you have suggested.
Point 2: First, I do not fully agree with the first sentence of the conclusions, namely “The present study answered the question of how the spread of organic farming could be accelerated, by focusing on the determinants of farmers’ organic conversion decisions”. The article doesn’t answer this question, it only reviews the factors that influence farmers’ decisions, but it doesn’t evaluate the significance of each factor. The article provides a review and synthesis of possible factors.
Response 2: Thank you very much for your comment. We have changed the sentence to: “The present study helps to answer the question of how the spread of organic farming…” (lines 746-747 & lines 9-10).
Let us also highlight: The study provides a review of factors that has been found to impact farmers’ decisions to convert conventional farming activities to organic activities, to help thus, in scheduling strategies that cause changes in these factors…
Point 3: Second, while it seems too late for the research process, I believe the applied approach and key words used to search for articles concerning the farmers’ decisions towards transition to organic agriculture was not selected in the best possible way. Word combinations could have been selected in a way to generate more search results, if only the authors have chosen more options. In this view the outputs of the Horizon 2020 LANDSUPPORT project (https://www.landsupport.eu) could be of use. Also, limiting conversion to only “organic” has narrowed it further, while close ideas of ecological farming covering low-input, conservative, integrated, agroecological have been omitted.
Response 3: Thanks the reviewer for the comments and suggestions! Our response is distinguished into three points that are next presented:
Point 3.1: Second, while it seems too late for the research process, I believe the applied approach and key words used to search for articles concerning the farmers’ decisions towards transition to organic agriculture was not selected in the best possible way. Word combinations could have been selected in a way to generate more search results, if only the authors have chosen more options.
Response 3.1: Some changes have been conducted in methodology section (lines 141-150 & 162 & 172-173). Let us also highlight that the focus of the research is the farmers’ decisions, intentions, attitudes, and perceptions related to organic conversion (lines 160-162).
Point 3.2: In this view the outputs of the Horizon 2020 LANDSUPPORT project (https://www.landsupport.eu) could be of use.
Response 3.2: Thank you. We found one published paper (reference 101) in Horizon 2020 LANDSUPPORT project that helped us. Let us also note that (lines 155-159), “...the first filtration was based both on the titles of the papers and whether they were published in peer-reviewed journals. A preference was indicated towards articles that were published in journals that have broad academic recognition in several scientific subjects, including the fields of agricultural economics, management, and marketing. In only a few cases were other scientific publications complementary used.”
Point 3.3: Also, limiting conversion to only “organic” has narrowed it further, while close ideas of ecological farming covering low-input, conservative, integrated, agroecological have been omitted.
Response 3.3: Thank you! Hope that the generalization and extension we highlight in the third paragraph of the conclusion (lines 764-777) can be viewed as a response to your valuable comment.
Point 4: Third, I would still suggest to look at the Horizon 2020 LIFT project’s results (https://www.lift-h2020.eu), as it is one of the key contributions on the subject at the moment and I’m honestly confused why the authors haven’t looked at it. It deals precisely with the decisions of the agricultural transformation towards ecological approaches and has been publishing outputs since 2018. The second project the outputs of which are omitted here is the Horizon 2020 UNISECO project (https://uniseco-project.eu), which also had made valuable contributions to agroecological farming theory and practice, gathering numerous outputs from stakeholders in regard to transition process. Both project have published reports and scientific articles.
Response 4: Thank you very much! We found three useful publications (references 102, 103, 104), from the Horizon 2020 UNISECO.
Point 5:Fourth, I am not sure the Figure 1 is the best way to depict the method applied.
Response 5: Thank you very much! The required changes have been made. The steps the Figure 1 showed has been described in the text (section 2, especially lines 141-150) while the figure is moved in Appendix A (lines 172-173 & lines 833-848).
Point 6: Fifth, there is an issue with Figure XX, as words at the bottom are only partially visible. Same with Table 1, here the issue is with the text on the left side.
Response 6: Thank you very much! As it is apparent in sections 2 and 3, the required corrections and improvements have been conducted: figure 1 (lines 203-220) & table 1 (lines 229-232).
Point 7: Sixth, a professional English proof-reading is needed, as the style and grammar are not polished yet.
Response 7: A language editing has been conducted by an MDPI expert. Thank you very much!
This manuscript is a resubmission of an earlier submission. The following is a list of the peer review reports and author responses from that submission.
Round 1
Reviewer 1 Report
The paper contains interesting results of literature review focused on organic farming with special consideration of factors influencing farmers’ decisions on conversion into its methods. In such sense it could inspire continuation of research at the matter. The paper is well written, English is acceptable however it should be improved in some fragments or sentences. The text should be revised according to the review because in the presented version it does not fulfill the standards of a scientific article:
Line 8: Abstract: “The share of the organic farming remains too low until now, though the organic conversion enables the food consumption and production to become sustainable”. The share in what?
The abstract does not include any conclusions or results of the study. In the lines 19-22 it is written that: “The importance of factors that impact decisions differs between enterprises/industries, locations, regions and countries. Public policies and private strategies can effectively accelerate the organic conversion by scheduling changes on factors which are important in each case.” We do not know which policies and strategies can effectively accelerate the organic conversion. We also do not know the “implications for practitioners and for future research”.
The keywords include “acceleration” and “factors”. Acceleration of what? Which factors?
In the introduction the authors cite a lot of papers connected with organic farming but there is no research gap identified in that section. The line 28 – the authors wrote that: “The environmental footprint of the Europe is much higher than the global average, while the agriculture is blamed that is a key source of environmental pressures.” There are no data on the difference between environmental footprint of the Europe and the global average. The aim of the paper is properly formulated but the content of particular sections of its structure should be improved. The part in which we should have discussion is not related to the scientific studies cited in the introduction.
The authors wrote that the identification of factors that impact the organic conversion decisions is based on scientific studies that have been published mostly in peer reviewed journals. In fact it is the only method used in the elaboration. The fragment presented in lines 124-146 is not related to methodology used in the paper. That content could be included in introduction and discussed in the further sections but it should be excluded from the section “methodology”.
The lines 151-152: “Because there are differences between countries in regards the share of organic farming and food consumption [15], in a first step we search for market, institutional, social and policy specific factors which can determine farmers' decisions.” It is hard to understand why do the authors search for these factors? “Because of differences between countries”? It is not good logic explanation why the Authors choose these factors to investigate. Without these differences there would not be any reasons to search these factors? There is the need for scientific justification why these particular factors are described in the paper. It refers both to external and internal factors as well as to the methods used to create classification presented in the figure 1.
Generally, methodology is very poor in the elaboration. We can only read, that the authors sought papers that examine different group of factors influencing conversion into organic farming and, that they described them in two groups (external and internal ones). Finally, they tried to build a framework which includes the sets of factors identified worldwide. The review of available literature is the only method in the study. It is not enough in scientific paper. Even if the text is based on published papers, the authors should indicate methods used to achieve the goal of the paper. Is it a descriptive analysis or is it comparative analysis? They could be used as the introduction to more sophisticated classification and for creation of model presenting interrelation between different factors of farmers’ decisions. However, the methods used to achieve these results should be described first. Besides, we do not know any criteria of the choice of particular factors taken into consideration and why they were grouped in the way presented in the elaboration.
The sections 3 “External farm business environment” does not include any statistical data connected to the most of presented factors. For example – there are no data on the value of the demand, its change or share in global demand for food. It could refer to the global market or market in the EU or in the chosen countries.
Chosen factors influencing conversion into organic faming are discussed in the section 3, but there are no binding outcomes from these characteristics. Furthermore, some fragments lacks of necessary explanations. For example, when authors write about public policy they state, that:”The decisions of producers involve several types of complex motivations. The most important motives include both economic and environmental considerations, followed by social motivations.” There is no explanation which economic, environmental and social motivations are involved in the decisions and how they impact these decisions. “Some policy regulations motivate higher organic adoption rates (…)” There is nothing about these regulations in the text. Finally, there is the need to indicate the general mechanisms (and areas) of public policy which can stimulate conversion or can create barriers to it. Similar mechanisms of the influence should be described in relation to every discussed factor.
Conclusion and propositions. There are no valuable conclusions and remarks presented in that section. Two first paragraphs (lines 441-475) are rather the summary of the previous sections. There are some obvious sentences without any results from the elaboration:
Lines 476-477 “The conversion of conventional farming to organic can be accelerated if the properly chosen changes are made in certain important factors that impact farmers’ decisions, intentions and attitudes which are related to the organic farming.” We do not know these “properly chosen changes”. Which changes are needed? They should be indicated as the results of the sections of the study previously presented in the text.
Lines 478-479. “Such changes can be effectively caused by the proper engagement of the public authorities and private actors who interact, in combination with the institutional arrangements [51,52,74].” What is the “proper engagement of the public authorities and private actors”? It should be specified.
Lines 480-482. “More specifically, the farm business environment can become more conducive for organic conversion by the properly designed interventions of the public authorities as well as by the strategies and actions of the processors and marketers of agricultural”. Which “properly designed interventions of the public authorities as well as by the strategies and actions of the processors and marketers” should be implemented and why? It should be written.
Lines 484-485. “The public policies, private actors’ and non-governmental organizations’ strategies can effectively accelerate the organic conversion decisions”. How can they accelerate these decisions? Which actions can be undertaken?
Line 485 “if they take into account a large number of factors, which determine the farmers’ decisions”. Which factors should be taken into account? What should be done if they were taken into account?
The answers could be included in the section 5.2. Propositions.
One answer is presented in the lines 498-500: “the public authorities and private actors must not neglect the provision of education to farmers, the dissemination of new knowledge and technology as well as the provision of individually customized information.”
The second answer is in the lines 505-508: “the public policies and the actions of the market participants should be also directed to changes, such as the consumers’ behaviour towards organic products, the access, efficiency and transparency of organic markets, the effectiveness of organic supply chains as well as the cost of organic conversion and certification [33,41,42,74,76,80,81].”. These answers are too general and not enough. They do not indicate measures, instruments actions institutions or organisations which could help do develop organic faming or enhance farmers to make decisions on conversion into organic farming.
In the lines 518-519 the authors indicate the research gap: “the insufficiency of the information provided to the public authorities and private actors that aim to effectively accelerate the conversion of conventional farming into organic”. It could be presented in the introduction as the argument for the discussion in the paper. The text does not include any results connected with that gap.
Author Response
Response to Reviewer 1 Comments
Point 1: The paper contains interesting results of literature review focused on organic farming with special consideration of factors influencing farmers’ decisions on conversion into its methods. In such sense it could inspire continuation of research at the matter. The paper is well written, English is acceptable however it should be improved in some fragments or sentences. The text should be revised according to the review because in the presented version it does not fulfill the standards of a scientific article:
Response 1: Thank you for your valuable comments and suggestions!
We have substantially reworked the paper taking into consideration all comments, point by point. After the elaboration of the manuscript, a language editing has been conducted.
All changes are highlighted within the attached document by using the track changes mode in MS Word.
Our response is also reported next, point by point.
Point 2: Line 8: Abstract: “The share of the organic farming remains too low until now, though the organic conversion enables the food consumption and production to become sustainable”. The share in what?
The abstract does not include any conclusions or results of the study. In the lines 19-22 it is written that: “The importance of factors that impact decisions differs between enterprises/industries, locations, regions and countries. Public policies and private strategies can effectively accelerate the organic conversion by scheduling changes on factors which are important in each case.” We do not know which policies and strategies can effectively accelerate the organic conversion. We also do not know the “implications for practitioners and for future research”.
Response 2: Thank you for your valuable comment.
We have elaborated on the abstract. We hope our changes are satisfactory and in line with what the reviewer suggests.
Point 3:The keywords include “acceleration” and “factors”. Acceleration of what? Which factors?
Response 3: We thank the reviewer for the comment.
Some changes have been done, following reviewer’s comments.
Point 4: In the introduction the authors cite a lot of papers connected with organic farming but there is no research gap identified in that section. The line 28 – the authors wrote that: “The environmental footprint of the Europe is much higher than the global average, while the agriculture is blamed that is a key source of environmental pressures.” There are no data on the difference between environmental footprint of the Europe and the global average. The aim of the paper is properly formulated but the content of particular sections of its structure should be improved. The part in which we should have discussion is not related to the scientific studies cited in the introduction.
Response 4: We thank the reviewer for the valuable comments.
We have tried to substantially improve the introduction. We have clearly present research gap and the main goal of the manuscript, as well as we have elaborated on the 3 and 4 sections and introduced a discussion section. We hope our changes are satisfactory and in line with what the reviewer suggests.
Point 5: The authors wrote that the identification of factors that impact the organic conversion decisions is based on scientific studies that have been published mostly in peer reviewed journals. In fact it is the only method used in the elaboration. The fragment presented in lines 124-146 is not related to methodology used in the paper. That content could be included in introduction and discussed in the further sections but it should be excluded from the section “methodology”.
Response 5: We thank the reviewer for the valuable comments and the suggestion.
We have elaborated on the methodology section and we have made adaptations in introduction, following reviewer’s comments.
Point 6: The lines 151-152: “Because there are differences between countries in regards the share of organic farming and food consumption [15], in a first step we search for market, institutional, social and policy specific factors which can determine farmers' decisions.” It is hard to understand why do the authors search for these factors? “Because of differences between countries”? It is not good logic explanation why the Authors choose these factors to investigate. Without these differences there would not be any reasons to search these factors? There is the need for scientific justification why these particular factors are described in the paper. It refers both to external and internal factors as well as to the methods used to create classification presented in the figure 1.
Response 6: We thank the reviewer for the valuable comments.
We have substantially elaborated on all sections of the manuscript, doing the improvements the reviewer suggests. More precisely, we tried to define the goal of the article and clearly present the methodology, attempting to eliminate phrases and reports that nothing add to the manuscript, while confuse the reader. We have also tried to make sufficiently apparent the classification of the factors. For example, some definitions are given in the methodology, as well as at the start of the sections 3 and 4.
Point 7: Generally, methodology is very poor in the elaboration. We can only read, that the authors sought papers that examine different group of factors influencing conversion into organic farming and, that they described them in two groups (external and internal ones). Finally, they tried to build a framework which includes the sets of factors identified worldwide. The review of available literature is the only method in the study. It is not enough in scientific paper. Even if the text is based on published papers, the authors should indicate methods used to achieve the goal of the paper. Is it a descriptive analysis or is it comparative analysis? They could be used as the introduction to more sophisticated classification and for creation of model presenting interrelation between different factors of farmers’ decisions. However, the methods used to achieve these results should be described first. Besides, we do not know any criteria of the choice of particular factors taken into consideration and why they were grouped in the way presented in the elaboration.
Response 7: We thank the reviewer for the valuable comments and suggestions.
We have elaborated on the methodology section, following all reviewer's comments. Le us highlight that the article is an integrative review (Torraco 2016). We hope our changes are satisfactory and in line with what the reviewer suggests.
Point 8: The sections 3 “External farm business environment” does not include any statistical data connected to the most of presented factors. For example – there are no data on the value of the demand, its change or share in global demand for food. It could refer to the global market or market in the EU or in the chosen countries.
Chosen factors influencing conversion into organic faming are discussed in the section 3, but there are no binding outcomes from these characteristics. Furthermore, some fragments lacks of necessary explanations. For example, when authors write about public policy they state, that:”The decisions of producers involve several types of complex motivations. The most important motives include both economic and environmental considerations, followed by social motivations.” There is no explanation which economic, environmental and social motivations are involved in the decisions and how they impact these decisions. “Some policy regulations motivate higher organic adoption rates (…)” There is nothing about these regulations in the text. Finally, there is the need to indicate the general mechanisms (and areas) of public policy which can stimulate conversion or can create barriers to it. Similar mechanisms of the influence should be described in relation to every discussed factor.
Response 8: We thank the reviewer for the valuable comments and suggestions.
We have introduced data where it is necessary, while we attempt to defined the concepts and phrases the reviewer highlights, as well as to reduce some ambiguities that arise.
Any information and report that nothing adds to the manuscript, has been eliminated. We have tried to make improvements in all places of the manuscript, as it is apparent by the “track changes”.
We hope our definitions and changes are satisfactory and in line with what the reviewer suggests.
Point 9: Conclusion and propositions. There are no valuable conclusions and remarks presented in that section. Two first paragraphs (lines 441-475) are rather the summary of the previous sections. There are some obvious sentences without any results from the elaboration:
Lines 476-477 “The conversion of conventional farming to organic can be accelerated if the properly chosen changes are made in certain important factors that impact farmers’ decisions, intentions and attitudes which are related to the organic farming.” We do not know these “properly chosen changes”. Which changes are needed? They should be indicated as the results of the sections of the study previously presented in the text.
Lines 478-479. “Such changes can be effectively caused by the proper engagement of the public authorities and private actors who interact, in combination with the institutional arrangements [51,52,74].” What is the “proper engagement of the public authorities and private actors”? It should be specified.
Lines 480-482. “More specifically, the farm business environment can become more conducive for organic conversion by the properly designed interventions of the public authorities as well as by the strategies and actions of the processors and marketers of agricultural”. Which “properly designed interventions of the public authorities as well as by the strategies and actions of the processors and marketers” should be implemented and why? It should be written.
Lines 484-485. “The public policies, private actors’ and non-governmental organizations’ strategies can effectively accelerate the organic conversion decisions”. How can they accelerate these decisions? Which actions can be undertaken?
Line 485 “if they take into account a large number of factors, which determine the farmers’ decisions”. Which factors should be taken into account? What should be done if they were taken into account?
The answers could be included in the section 5.2. Propositions.
Response 9: We thank the reviewer for the valuable comments and suggestions.
We have totally elaborated on the manuscript, attempting to give as more as possible clarifications and examples, in all sections, following all the reviewer’s comments and recommendations, while we have added a discussion section.
We attempt to reshape many points in the manuscript, and give sufficient definitions and detailed explanations for phrases and concepts the reviewer highlights, such as: “properly chosen changes”, “engagement of the public authorities and private actors”, “properly designed interventions of the public authorities as well as by the strategies and actions of the processors and marketers”, “actions should be undertaken”, "factors should be taken into account", "policy mechanisms".
We have also eliminated some phrases and notes that nothing add to the manuscript.
All changes are highlighted within the document by using the "track changes" mode.
We hope our definitions and changes are satisfactory and in line with what the reviewer suggests.
Point 10: One answer is presented in the lines 498-500: “the public authorities and private actors must not neglect the provision of education to farmers, the dissemination of new knowledge and technology as well as the provision of individually customized information.”
The second answer is in the lines 505-508: “the public policies and the actions of the market participants should be also directed to changes, such as the consumers’ behaviour towards organic products, the access, efficiency and transparency of organic markets, the effectiveness of organic supply chains as well as the cost of organic conversion and certification [33,41,42,74,76,80,81].”. These answers are too general and not enough. They do not indicate measures, instruments actions institutions or organisations which could help do develop organic faming or enhance farmers to make decisions on conversion into organic farming.
In the lines 518-519 the authors indicate the research gap: “the insufficiency of the information provided to the public authorities and private actors that aim to effectively accelerate the conversion of conventional farming into organic”. It could be presented in the introduction as the argument for the discussion in the paper. The text does not include any results connected with that gap.
Response 10: We thank the reviewer for the valuable comments and suggestions.
We have elaborated on the conclusion and propositions section, while we have added the discussion section, taking into consideration all the reviewers’ comments.
In regards the comment in the lines 518-519, it has been incorporated in the introduction section.
As it is previously noted, we have tried to make improvements in all places of the manuscript, that are apparent by the “track changes”.
We hope that we sufficiently have responded to the reviewer’s comments and recommendations.

Reviewer 2 Report
The paper has an ambitious objective, it wants to review and synthesize the factors that determine adoption of organic farming principles. Yet, the method, materials used and the formal analysis are not well presented and in my opinion are also not rigorous enough to back this ambitious objective.
A lot of the content about adoption factors is common sense, but presented in a generalized manner which lacks consideration of the true heterogeneity of situations and contexts in which farmers convert to organic farming. Often, the authors dive into aspects that are absolutely not relevant for the topic (see for instance the excursus on why ICT matters for agriculture). On the other hand, they do not explain well what the underlying data has been for their analysis and how this literature was precisely identified (keywords, etc). They mention a lot of databases, but a rigorous review requires 1-2 (scopus and/or web of science), but then in a well conducted manner. Of course, literature is cited, but this paper would benefit, if a better overview is provided on what literature is available and analyzed with respect to each of the factor. For instance, the paragraph on ICT has no connection to organic farming whatsoever.
While the abstract is okay, already the first paragraph is written in poor English!
I highly recommend the authors to have their manuscript checked by a native speaker and to better trim down to the essentials. The scope is unclear and the clarity suffers from that.
Some specific comments
Memken and Qaim, 2018 provide a good meta analysis of organic benefits.
Line 57-59, which source do you have for this?
Line 74-77 better provide annualized growth (compounded annual growth rates)
Line 84 use consistent decimal separators
Line 97: It is hard to imagine that no study to some extent has done a comprehensive analysis of organic farming adoption factors (see also studies cited in line 147). You should also mention those studies that did something similar but not exactly the same.
Line 110-15: may skip this paragraph
Methodology: It is unclear how many sources (# peer reviewed # grey literature, etc) constitute the underlying data of the analysis
The approach by Freund-Lüdeke may be fine, but where has it been applied? An
Figure 1 is not formatted well, some text is covered
- 217 – distance to market is very dependent on the type of market and the type of agricultural good. Take the example of organic pineapples shipped from West Africa to the EU, the market is far away but the product is not that perishable. Your discussion here is not very informative neither does it sufficiently reflect the heterogeneity of such cases. I fear this is a consistent problem of this article.
For instance: it would be good if the authors provided some sort of simple quantitative analysis: How many studies concluded distance to markets was an adoption barrier and how many didn’t?
L 263-267: I am confused: What do you want to say here?
L 275: The whole paragraph on ICT fails your objective to write about the factors that determine adoption of organic farming. Either rewrite and connect to organic or omit.
Section on Institutional settings and Public Policy => see different settings (bottom up or top down, the latter was analysed for the case of Bhutan by Feuerbacher et al. 2017)
Section 3.6: Economic factors – this is very brief. No mentioning of the role of yield gaps (and thus lower productivity) at all. see Seufert et al 2012 for a review.
Author Response
Response to Reviewer 2 Comments
Point 1: The paper has an ambitious objective, it wants to review and synthesize the factors that determine adoption of organic farming principles. Yet, the method, materials used and the formal analysis are not well presented and in my opinion are also not rigorous enough to back this ambitious objective.
Response 1: Thank you for your valuable comments!
We have substantially elaborated on the paper taking into consideration all comments, point by point. After the elaboration of the manuscript, a language editing has been conducted.
All changes are highlighted within the attached document by using the track changes mode in MS Word.
Our response is also reported next, point by point.
Point 2: A lot of the content about adoption factors is common sense, but presented in a generalized manner which lacks consideration of the true heterogeneity of situations and contexts in which farmers convert to organic farming. Often, the authors dive into aspects that are absolutely not relevant for the topic (see for instance the excursus on why ICT matters for agriculture). On the other hand, they do not explain well what the underlying data has been for their analysis and how this literature was precisely identified (keywords, etc). They mention a lot of databases, but a rigorous review requires 1-2 (scopus and/or web of science), but then in a well conducted manner. Of course, literature is cited, but this paper would benefit, if a better overview is provided on what literature is available and analyzed with respect to each of the factor. For instance, the paragraph on ICT has no connection to organic farming whatsoever.
Response 2: We thank the reviewer for the valuable comments and the suggestion.
We have totally elaborated the methodology section and we have made adaptations and provided justifications, both in introduction and discussion sections, following reviewer’s comments.
In regards the paragraph on ICT, let us to note: Because the use of ICT in food supply is rapidly developed last years, becoming it a promising issue, we choose to rely on selected literature to understand the expected impact of available ICT on farmers’ organic conversion decisions. We take also into account that previous research reveals that the use of ICT is a determinant of farmers’ organic conversion decisions.
Point 3:While the abstract is okay, already the first paragraph is written in poor English!
Response 3: Thank the reviewer!
A language editing has been conducted by an MDPI expert.
Point 4: I highly recommend the authors to have their manuscript checked by a native speaker and to better trim down to the essentials. The scope is unclear and the clarity suffers from that.
Response 4: We thank the reviewer for the valuable comments and suggestions.
We have elaborated on all sections, following the reviewer's comments and suggestions. More precisely, we clearly define the goal of the article and the methodology, we have improved the two sections that include the decisions determinants, as well as we have added a discussion section.
After the elaboration of the manuscript, a language editing has been conducted by a professional.
Point 5:
Memken and Qaim, 2018 provide a good meta analysis of organic benefits.
Line 57-59, which source do you have for this?
Line 74-77 better provide annualized growth (compounded annual growth rates)
Line 84 use consistent decimal separators
Line 97: It is hard to imagine that no study to some extent has done a comprehensive analysis of organic farming adoption factors (see also studies cited in line 147). You should also mention those studies that did something similar but not exactly the same.
Line 110-15: may skip this paragraph
Methodology: It is unclear how many sources (# peer reviewed # grey literature, etc) constitute the underlying data of the analysis
The approach by Freund-Lüdeke may be fine, but where has it been applied? An
Figure 1 is not formatted well, some text is covered
Response 5: We thank the reviewer for the valuable comments and the suggestion.
As it is apparent in the highlighted points, we have elaborated all sections of the manuscript, following all reviewer’s comments and suggestions.
In regards the line 97: Sufficient definitions are given in methodology section. It could be useful to note that “we focus on those factors for which it is found to impact farmers’ organic conversion decisions”.
In regards "The approach by Freund-Lüdeke:" We highlight now that we are borrowed ideas from the sustainable business model the study suggests, to understand and organize the farmers’ organic decision process. Thus, it might not help much if we mentioned applications of the model, while they may have made the article more complicated.
Point 6:
L 217 – distance to market is very dependent on the type of market and the type of agricultural good. Take the example of organic pineapples shipped from West Africa to the EU, the market is far away but the product is not that perishable. Your discussion here is not very informative neither does it sufficiently reflect the heterogeneity of such cases. I fear this is a consistent problem of this article.
For instance: it would be good if the authors provided some sort of simple quantitative analysis: How many studies concluded distance to markets was an adoption barrier and how many didn’t?
Response 6: We thank the reviewer for the valuable comments and suggestions.
We hope that sufficient definitions, details and documentations are given now, for the distance, while some indicative quantitative data are also provided in some points. A quantitative analysis goes beyond of the main goal of the study. It could be possible to be conducted in a future study.
We hope our changes are satisfactory and in line with what the reviewer suggests.
Point 7:
L 263-267: I am confused: What do you want to say here?
L 275: The whole paragraph on ICT fails your objective to write about the factors that determine adoption of organic farming. Either rewrite and connect to organic or omit.
Section on Institutional settings and Public Policy => see different settings (bottom up or top down, the latter was analysed for the case of Bhutan by Feuerbacher et al. 2017)
Section 3.6: Economic factors – this is very brief. No mentioning of the role of yield gaps (and thus lower productivity) at all. see Seufert et al 2012 for a review.
Response 7: We thank the reviewer for the valuable comments and suggestions.
We have elaborated on the manuscript including also the points the reviewer suggests. In regards the set of economic factors we change it to financial factors. We hope our changes are satisfactory and in line with what the reviewer suggests.

Reviewer 3 Report
The authors made concerted effort to offer a comprehensive review on the factors that impact organic conversion decisions. The paper aims to identify, bring together and systematically present the factors that impact the organic conversion decisions of farmers worldwide. The thought process for organizing the article is good and presented in a logical sequence. However, there are ample rooms for improvement in the paper. Some of these are below:
- The paper has substantial information and well-enough logical organization. However, those are not presented with substantial brevity, rather are descriptions with little depth of critical analysis. The authors need to re-read the manuscript and add critical analysis of the argument they are pursuing. They also need to current spellings [not many] and grammatical errors [or word omissions or inclusions].
- In many places, proper proof reading and editorial changes are necessary, without which the reader will not find the logical progression / development of ideas. For example, in lines 34-35, “In regards to organic farming, it has been revealed that has a lower footprint with a higher ……” There are several similar ambiguous presentations throughout the paper.
- The division of Introduction section into two subsections came abruptly. The development of ideas to the objectives of the paper needs to be flushed out.
- A review article is expected to be a critical review, not merely a description of who did what. There are places, where the ‘why’ or ‘how’ question becomes obvious. For example, how organic farming exhibits lowers emission and higher carbon sequestration [lines 39-40].
- The sentence on line 57-59 does not seem a concluding sentence as it is not entirely derived from the sentences in the paragraph. If not, a reference is warranted.
- Is the health impact of organic is due to food habit or lifestyle? They are not identical. Lifestyle is much broader than food habit. However, a change in food habit may contribute to change in lifestyle and vice-versa.
- Line 128, a reader will be wandering what is ICT?
- The figure needs to be revised to present the logical sequence. The one-way contribution of external factors and reversible contribution of internal factors are understandable, but those need to be drawn in a way that one is not preferred or more valuable over the other. Not sure, what the ‘delta’ sign means on the right side of the Outcome box. Part of the market factors got hidden.
- In the last paragraph under proposition, the authors bring the corona virus issue. The scope of the review is already overly broad and nowhere corona virus has been brought. I would recommend dropping the last paragraph altogether.
Author Response
Response to Reviewer 3 Comments
Point 1: The authors made concerted effort to offer a comprehensive review on the factors that impact organic conversion decisions. The paper aims to identify, bring together and systematically present the factors that impact the organic conversion decisions of farmers worldwide. The thought process for organizing the article is good and presented in a logical sequence. However, there are ample rooms for improvement in the paper. Some of these are below:
Response 1: Thank you for your valuable comments! We have substantially reworked the paper taking into consideration all comments point by point.
All changes are highlighted within the attached document by using the track changes mode in MS Word.
Our response is also reported next, point by point.
Point 2: The paper has substantial information and well-enough logical organization. However, those are not presented with substantial brevity, rather are descriptions with little depth of critical analysis. The authors need to re-read the manuscript and add critical analysis of the argument they are pursuing. They also need to current spellings [not many] and grammatical errors [or word omissions or inclusions].
In many places, proper proof reading and editorial changes are necessary, without which the reader will not find the logical progression / development of ideas. For example, in lines 34-35, “In regards to organic farming, it has been revealed that has a lower footprint with a higher ……” There are several similar ambiguous presentations throughout the paper.
Response 2: Thank you for your valuable comments and suggestions.
We have elaborated all sections of the manuscript, following reviewer’s comments and suggestions. A discussion section has been added, while after the elaboration of the manuscript, a language editing has been conducted by a professional expert. We hope our changes are satisfactory and in line with what the reviewer suggests.
Point 3: The division of Introduction section into two subsections came abruptly. The development of ideas to the objectives of the paper needs to be flushed out.
Response 3: We thank the reviewer for the comment.
As it is apparent by the “track changes”, many improvements have been done following the reviewer’s comments and suggestions. We have tried to substantially improve the introduction. We hope that we have clearly presented the research gap and the main goal of the manuscript, as well as that the introduction has sufficiently been improved.
Point 4: A review article is expected to be a critical review, not merely a description of who did what. There are places, where the ‘why’ or ‘how’ question becomes obvious. For example, how organic farming exhibits lowers emission and higher carbon sequestration [lines 39-40].
Response 4: We thank the reviewer for the valuable comments.
After we have clearly presented the main goal of the manuscript, we have totally elaborated on the methodology. It is highlighted that the article is an integrative review (Torraco 2016). We have tried to make improvements in all places of the manuscript, as it is apparent by the “track changes”. We hope our changes are satisfactory and in line with what the reviewer suggests.
Point 5: The sentence on line 57-59 does not seem a concluding sentence as it is not entirely derived from the sentences in the paragraph. If not, a reference is warranted.
Response 5: We thank the reviewer for the valuable comment and the suggestion.
The sentence is a continuation of the bibliographic references [9,12].
Point 6: Is the health impact of organic is due to food habit or lifestyle? They are not identical. Lifestyle is much broader than food habit. However, a change in food habit may contribute to change in lifestyle and vice-versa.
Response 6: We thank the reviewer for the valuable comments.
We have made the required merge and improvement.
Point 7: Line 128, a reader will be wandering what is ICT?
Response 7: We thank the reviewer for the valuable comment.
We have properly defined it, following journal guidelines.
Point 8: The figure needs to be revised to present the logical sequence. The one-way contribution of external factors and reversible contribution of internal factors are understandable, but those need to be drawn in a way that one is not preferred or more valuable over the other. Not sure, what the ‘delta’ sign means on the right side of the Outcome box. Part of the market factors got hidden.
Response 8: We thank the reviewer for the valuable comments and suggestions.
We have made all the improvements required, following reviewer’s comments in figure. In regards the direction of the impact of factors, useful definitions are also given in the start of the two sections: 3 and 4. We hope our changes are satisfactory and in line with what the reviewer suggests.
Point 9: In the last paragraph under proposition, the authors bring the corona virus issue. The scope of the review is already overly broad and nowhere corona virus has been brought. I would recommend dropping the last paragraph altogether.
Response 9: We thank the reviewer for the valuable comment and suggestion.
This paragraph has been eliminated as the reviewer suggests.

Reviewer 4 Report
The review presents an interesting and important topic covering factors that affect farmers’ decision regarding organic conversion decisions.
Introduction is adequate and sufficient.
Methodology is not sufficiently presented, as it presents sources of information (articles), but doesn’t quantify how many articles from these sources (databases) were covered in the research process. If these were only the 83 references presented then the declaration of eight major databases doesn’t seem to be adequate. A quick search reveals several articles that were not covered in the presented research and could be a valuable contribution. Therefore the question stands to what extent does the authors’ statement “systematic search has been performed” is true, or when was it exactly conducted.
Grouping of decision factors doesn’t seem to be clear enough and is up to discussion. For example, while the market factors are elaborated and analyzed step by step, economic factors are basically just mentioned as some general idea. The difference between the market and economic factors in authors’ understanding is not clear. Information and communication technology (ICT) are not present in the Figure 1, as well as information provision is not visible there.
The discussion is not present in the article and would be required to understand how the authors’ approach to classification of factors differs from those of other researchers.
The value added of the article is not clear, basically it is an aggregation of various approaches used in the literature, yet the differences in approaches to classification of conversion factors are not reflected. If the result of the article is solely the synthesis (which is of course valuable as well, since the manuscript is a review), this synthesis should be better structured and the different approaches to outlining the factor groups should be presented. Otherwise it is not clear if the proposed by the authors’ factor groups are doing a better job to factor structuring.
I would suggest to use the materials developed by the LIFT project (www.lift-h2020.eu) as it covers exactly the topic researched in the article. While I would suggest to look through its public reports found at https://www.lift-h2020.eu/deliverables, also one of the most recent publications could be useful: Duvaleix, S.; Lassalas, M.; Latruffe, L.; Konstantidelli, V.; Tzouramani, I. Adopting Environmentally Friendly Farming Practices and the Role of Quality Labels and Producer Organisations: A Qualitative Analysis Based on Two European Case Studies. Sustainability 2020, 12, 10457, https://www.mdpi.com/2071-1050/12/24/10457.
Many parts of the text in Figure 1 is not visible.
Line 207 what does the number “30” in the brackets refer to (possibly reference, but must be fixed).
Substantial English editing is needed, there are quite many errors: orthographic (spelling and punctuation), grammatical and stylistic, yet a professional proofreading could solve this problem.
Author Response
Response to Reviewer 4 Comments
Point 1: The review presents an interesting and important topic covering factors that affect farmers’ decision regarding organic conversion decisions.
Response 1: Thank you for your valuable comments! We have elaborated on the paper taking into consideration all comments point by point.
All changes are highlighted within the attached document by using the track changes mode in MS Word.
Our response is also reported next, point by point.
Point 2: Introduction is adequate and sufficient.
Response 2: Thank you.
Point 3: Methodology is not sufficiently presented, as it presents sources of information (articles), but doesn’t quantify how many articles from these sources (databases) were covered in the research process. If these were only the 83 references presented then the declaration of eight major databases doesn’t seem to be adequate. A quick search reveals several articles that were not covered in the presented research and could be a valuable contribution. Therefore the question stands to what extent does the authors’ statement “systematic search has been performed” is true, or when was it exactly conducted.
Response 3: We thank the reviewer for the comment.
We have elaborated on the methodology section, following reviewer’s comments. We have also added some new references in the manuscript, mostly focusing on studies that explore the factors that determine farmers’ organic conversion decisions.
Point 4: Grouping of decision factors doesn’t seem to be clear enough and is up to discussion. For example, while the market factors are elaborated and analyzed step by step, economic factors are basically just mentioned as some general idea. The difference between the market and economic factors in authors’ understanding is not clear. Information and communication technology (ICT) are not present in the Figure 1, as well as information provision is not visible there.
Response 4: We thank the reviewer for the valuable comments.
We have tried to substantially improve the manuscript. We have clearly present research gap and the main goal of the manuscript. We clearly explained the grouping of factors at the end of the methodology and at the start of sections 3, 4 and 5, while we have changed the set of economic factors to financial factors. We have also elaborated on the figure 1. We hope our changes are satisfactory and in line with what the reviewer suggests.
Point 5: The discussion is not present in the article and would be required to understand how the authors’ approach to classification of factors differs from those of other researchers.
Response 5: We thank the reviewer for the valuable comment and the suggestion.
We have introduced a discussion section, following reviewer’s comments.
Point 6: The value added of the article is not clear, basically it is an aggregation of various approaches used in the literature, yet the differences in approaches to classification of conversion factors are not reflected. If the result of the article is solely the synthesis (which is of course valuable as well, since the manuscript is a review), this synthesis should be better structured and the different approaches to outlining the factor groups should be presented. Otherwise it is not clear if the proposed by the authors’ factor groups are doing a better job to factor structuring.
Response 6: We thank the reviewer for the valuable comments.
We have substantially elaborated all sections, doing the improvements the reviewer suggests. More precisely, we have tried to make apparent the value added of the manuscript, we clearly define the goal of the article and sufficiently described the methodology, following reviewer’s comments. Because the article is an integrative review (Torraco’s 2016), selected studies are cited as examples of results illustrating the article’s argument, which is linked with the determinants of the decisions of farmers to convert the conventional farming activities to organic. We have also tried to make sufficiently apparent the classification of the factors. We hope our changes are satisfactory and in line with what the reviewer suggests.
Point 7: I would suggest to use the materials developed by the LIFT project (www.lift-h2020.eu) as it covers exactly the topic researched in the article. While I would suggest to look through its public reports found at https://www.lift-h2020.eu/deliverables, also one of the most recent publications could be useful: Duvaleix, S.; Lassalas, M.; Latruffe, L.; Konstantidelli, V.; Tzouramani, I. Adopting Environmentally Friendly Farming Practices and the Role of Quality Labels and Producer Organisations: A Qualitative Analysis Based on Two European Case Studies. Sustainability 2020, 12, 10457, https://www.mdpi.com/2071-1050/12/24/10457.
Response 7: We thank the reviewer for the valuable suggestions.
We have used the publication the reviewer suggest, while the materials developed by the LIFT project will be used in a future work. We hope our changes are satisfactory and in line with what the reviewer suggests.
Point 8: Many parts of the text in Figure 1 is not visible.
Response 8: We thank the reviewer for the valuable comment.
We have elaborated on the figure 1.
Point 9: Line 207 what does the number “30” in the brackets refer to (possibly reference, but must be fixed).
Response 9: We thank the reviewer for the valuable comment.
It is corrected.
Point 10: Substantial English editing is needed, there are quite many errors: orthographic (spelling and punctuation), grammatical and stylistic, yet a professional proofreading could solve this problem.
Response 10: We thank the reviewer for the valuable comments and suggestions.
After the elaboration of the manuscript, a language editing has been conducted by a professional expert.

Round 2
Reviewer 1 Report
I am afraid of that, the authors did not understand some of my comments from the first review. For example, I indicated wrong keywords like “acceleration”. They changed it into “determinants of acceleration”. I still do not know what the “acceleration” is. “Acceleration” of what? We also have “sets of factors” in the keywords. The factors of what?
Authors responded to Reviewer's Comments that they reported next their changes, point by point. They did not do it. They wrote that they improved some parts of the article, but they did not explain how they had done it in the authors’ response.
They wrote that they clearly presented the main goal of the manuscript. In the present form there is no goal formulated in the paper.
The authors changed the text, but they did not really improve its content according to the comments of the reviewer. They did not expand the methodology. The review of available literature is the only method in the study. It is not enough in scientific paper.
We still do not know any criteria of the choice of particular factors taken into consideration and why they were grouped in the way presented in the elaboration. It could be done according to previously established methods.
In the conclusion the authors wrote, that “the present study seeks to answer the question of how the spread of the organic farming could be accelerated”. Is it the goal of the study? If yes - it should be clearly presented in the introduction. The authors did not present any answer to that question. They summarized the presentation of the factors influencing farmers’ decisions but they did not explain which of them were driving forces or barriers for conversion into organic farming and why. There are no remarks concerning public activities which could accelerate that conversion.
Reviewer 4 Report
The article has been substantially modified with necessary elements outlined and emphasized upon. It is a valuable contribution and builds on both authors' vision and past works of other scientists.